# Structure and regulation of full-length human leucine-rich repeat kinase 1

Riley D. Metcalfe[1], Juliana A. Martinez Fiesco[1], Luis Bonet-Ponce [2], Jillian H. Kluss[2], Mark R. Cookson [2] & Ping Zhang [1] ✉

The human leucine-rich repeat kinases (LRRKs), LRRK1 and LRRK2 are large and unusually complex multi-domain kinases, which regulate fundamental cellular processes and have been implicated in human disease. Structures of LRRK2 have recently been determined, but the structure and molecular mechanisms regulating the activity of the LRRK1 as well as differences in the regulation of LRRK1 and LRRK2 remain unclear. Here, we report a cryo-EM structure of the LRRK1 monomer and a lower-resolution cryo-EM map of the LRRK1 dimer. The monomer structure, in which the kinase is in an inactive conformation, reveals key interdomain interfaces that control kinase activity as we validate experimentally. Both the LRRK1 monomer and dimer are structurally distinct compared to LRRK2. Overall, our results provide structural insights into the activation of the human LRRKs, which advance our understanding of their physiological and pathological roles.

Leucine-rich repeat kinase (LRRK) 1 is a multidomain protein which, near-uniquely, contains both a Ras-like GTPase domain and kinase domain in the same polypeptide chain. The two human LRRKs, LRRK1 and LRRK2, were first identified in 2002 in a genome screen for unidentified human kinases[1]. Shortly after, mutations in LRRK2 were found to be pathogenetic, causing both familial and sporadic Parkinson's disease (PD)[2,3]. This discovery prompted intensive investigation of the biological role of LRRK2, and the development of LRRK2-specific kinase inhibitors[4,5]. Despite this, the exact physiological roles of both LRRK1 and LRRK2 remain unclear. Both LRRK1 and LRRK2 phosphorylate distinct subsets of Rab GTPases on a conserved serine or threonine residue, thereby modulating interactions between the Rab proteins and their effectors[6].

The biological roles of LRRK1 are less well-established compared to LRRK2. LRRK1 and LRRK2 interact with distinct proteins and have different substrates, making them functionally non-redundant proteins[7,8]. LRRK1 does not phosphorylate the LRRK2 substrates Rab10 and Rab8. Instead, it phosphorylates Rab7, a key player in lysosomal biogenesis and trafficking[7,9]. Related to this, LRRK1 is involved in regulating mitophagy and autophagy[10,11]. Additionally, LRRK1 has a role in regulating the trafficking and lysosomal degradation of the epidermal growth factor receptor (EGFR)[12,13]. LRRK1 is itself tyrosine-phosphorylated by the EGFR, which places LRRK1 in a pathway regulating EGFR trafficking[13]. LRRK1 has a well-established role in bone development[14], with LRRK1 knockout mice exhibiting severe osteoporosis[15]. A genetic disease, osteosclerotic metaphyseal dysplasia (OSMD), which is characterized by severe bone abnormalities and osteoporosis, is caused by mutations in LRRK1 which eliminate LRRK1 kinase activity[7,16–18]. The Rho-family small GTPase Rac1, which has a role in regulating the actin cytoskeleton in osteoclasts[19], is phosphorylated by LRRK1[20], suggesting a mechanism for the link between the loss of LRRK1 activity and OSMD.

Both LRRK1 and LRRK2 are exceptionally large proteins, ranging from 230 to 280 kDa and containing 2000 to 2500 residues, with complex domain structures[21,22]. LRRK1 consists of N-terminal ankyrin repeats, leucine-rich repeats, and C-terminal Roc (Ras-of-complex) GTPase domain, COR (C-terminal of Roc) scaffolding domain, kinase domain and a WD40 repeat domain at the C-terminus. LRRK2 in addition to these domains, has armadillo repeats located N-terminal of the ankyrin repeats. Early biochemical analysis showed that both LRRKs are functional kinases, however biochemical characterization, and structural analysis was hindered by the size, flexibility, and low recombinant expression levels of both proteins[22–27]. Recently, cryo-electron microscopy (cryo-EM) structures have been reported of the

[1]Center for Structural Biology, Center for Cancer Research, National Cancer Institute, Frederick, MD 21702, USA. [2]Cell Biology and Gene Expression Section, Laboratory of Neurogenetics, National Institute on Aging, National Institutes of Health, Bethesda, MD 20892, USA. ✉e-mail: ping.zhang@nih.gov

catalytic C-terminal half of LRRK2[28], full-length inactive LRRK2[29], the catalytic half of LRRK2 engaged with microtubules[30] and the active LRRK2 bound to a Rab protein[31]. A low-resolution cryo-EM map of the catalytic half of LRRK1 suggests that the arrangement of the catalytic domains is conserved between LRRK1 and LRRK2[30], although a detailed structural analysis of the full-length LRRK1 has so far not been possible.

Here, we report the cryo-EM structure of the monomeric full-length LRRK1, and a lower-resolution map of the LRRK1 dimer. Our structures explain several regions known to regulate LRRK1 kinase activity. Additionally, they reveal notable structural differences between LRRK1 and LRRK2, particularly in the position and structural dynamics in the leucine-rich repeats, as well as interdomain contacts between the kinase and Roc domains. Our results provide a structural framework for investigating LRRK1 biology and enable the future exploration of unique and universal mechanisms involved in the activation and regulation of both LRRK1 and LRRK2.

## Results

### Purification and initial characterization of LRRK1

We purified full-length human LRRK1 (residues 1–2015) from baculovirus-infected insect cells, using sequential Flag-affinity and gel filtration chromatography (Supplementary Fig. 1A, B). The initial affinity purified protein eluted from the gel filtration column as a complex mixture of monomeric, dimeric, and larger species (Supplementary Fig. 1A, B). We determined the solution mass of the putative 'monomer' and 'dimer' fractions using mass photometry[32] (MP) (Fig. 1A), identifying species with masses consistent with the presence of the LRRK1 monomer (228 kDa) and dimer (448 kDa) in both fractions (Fig. 1A), with the monomer fraction predominately containing monomers, and the dimer fraction containing both monomers and dimers at approximately a 1:2 ratio, alongside other higher-order species. Furthermore, multi-angle light scattering (MALS) data collected on the monomer fraction confirmed a mass of 248 kDa, consistent with the expected mass of the LRRK1 monomer (Fig. 1B). These results demonstrate that LRRK1 exists as both a monomer and dimer in solution and does not preferentially form either species.

We observed robust Rab7A phosphorylation by the purified LRRK1 monomer using a Western blot assay (Supplementary Fig. 1C, D), confirming that the purified LRRK1 can phosphorylate its substrate.

### Structure of full-length human LRRK1

We subsequently undertook single particle cryo-EM of LRRK1 to determine the structure (Fig. 1C, D, Supplementary Fig. 2, see "Methods"). We collected data on grids prepared from both LRRK1 monomer and LRRK1 dimer fractions and processed the data together, as the LRRK1 dimer datasets contained a substantial number of LRRK1 monomer particles, as would be anticipated from the MP measurements (Fig. 1A, Supplementary Fig. 2A, B). 2D classification revealed several classes with the domains of LRRK1 clearly visible, and several classes with clear twofold symmetry, corresponding to the LRRK1 dimer (Supplementary Fig. 2A, B). Subsequent processing resulted in a 3.9 Å resolution map of the LRRK1 monomer and a 6.4 Å map of the LRRK1 dimer (Fig. 1C, D and Supplementary Figs. 2C, 3, 4). Local refinement improved the resolution of the C-terminal Roc-COR/ kinase/WD40 catalytic domains to 3.8 Å in the LRRK1 monomer map (Supplementary Figs. 2C and 3B). In the monomer map, large side-chains are visible and secondary-structure elements are well defined, α-helices clearly visible, and β-strands separated, consistent with a map reconstructed at this resolution (Supplementary Fig. 3E). However, the definition within the LRRs is notably poorer (Fig. 1Ci-ii, Supplementary Fig. 3A, B).

The overall structure of the LRRK1 monomer is 'O-shaped'. The structure is relatively compact, with the Roc, COR, kinase and WD40 domains arranged in a 'J-shape'. The leucine-rich repeats bridge the

Roc-COR and WD40 domains (Fig. 1C), although they do not form direct contacts with the WD40 domain. The ankyrin repeats are not well defined in the density, implying that they are flexible in the monomer, although poorly defined density N-terminal of the LRRs in the monomer map may correspond to the ankyrin repeats (Fig. 1Ci). We purified LRRK1 in the absence of any nucleotide and did not add any nucleotide prior to cryo-EM grid preparation. We did not observe any density for an adenosine nucleotide in the kinase active site. However, we included GDP in the atomic model as we did observe density for GDP in the Roc active site, likely representing endogenous GDP (Supplementary Fig. 5A, B). Reinforcing this observation, the Switch-I GTPase motif, which responds to guanosine nucleotide binding[33], is unambiguously in a conformation consistent with GDP binding (Supplementary Fig. 5A, B).

In the structure, the LRRK1 kinase domain is in an inactive conformation, and packs closely to the COR-B domain (Fig. 1C). The kinase regulatory (R)-spine[34] is broken, the αC helix is in the 'out' position, and the conserved K1270/E1307 ion pair is broken (between the αC helix and N-lobe, equivalent to the K72/E91 ion pair in PKA), all markers of an inactive kinase (Fig. 1E). The kinase activation loop is well-defined in the density, with the conserved DFG (DYG in LRRKs) residue D1409 in an 'out' conformation, pointed away from the ATP binding site.

The cryo-EM analysis of the LRRK1 dimer map reveals that two copies of the LRRK1 monomer can be accommodated within the dimer (Fig. 1Dii). Individual domains of the LRRK1 monomer are visible within the density (Fig. 1Dii). Additionally, we identified helical density, above the kinase domain in the map and adjacent to the LRRs, which correspond to the ankyrin repeats (Supplementary Fig. 4C). The ankyrin repeats form the dimerization interface between the two LRRK1 monomers. The LRRs and ankyrin repeats block the kinase active site of the adjacent LRRK1 molecule in the dimer, implying that the dimer is intrinsically inactive (Supplementary Fig. 4D), however the low resolution of the map prohibits a detailed analysis of the dimerization interface.

### Comparison of the structures of LRRK1 and LRRK2 reveal striking difference in position of LRRs

Recent structural studies of LRRK2 enable a comparison of experimental structures of LRRK1 and LRRK2[28–31] (Fig. 2). The most striking difference between the two proteins is in the position of the LRRs (Fig. 2A). In LRRK2, the LRRs and ANK repeats cover and occlude the kinase domain, presumably serving to regulate substrate access (Fig. 2A, B) with the additional N-terminal domains in LRRK2 projecting beyond the kinase-LRR 'core' of the protein, with the 'hinge helix' in the LRRs forming an interface between the ankyrin, LRR and WD40 domains[29]. The overall arrangement of the C-terminal catalytic domains is conserved between the two proteins, and the overall structure of the kinase domain in LRRK1 closely resembles that of the determined inactive LRRK2 kinase structures, with the major structural difference being an extended αC helix in LRRK1 (Fig. 2C, D). Relative to the kinase, the Roc domain is displaced by ~10 Å between the two proteins, which accommodates sterically the more compact position of the LRRs in LRRK2 (Fig. 2D). The curvature of the LRRs is very similar between the two proteins. The position of the LRRs in LRRK2 implies the requirement for rearrangement of the LRRs and N-terminal domains to permit substrate access to the kinase in LRRK2. In contrast, in LRRK1, the LRRs do not form interfaces with the remainder of the protein and do not occlude the kinase domain (Figs. 1 and 2B).

The kinase domain is occluded in both the LRRK1 and LRRK2 dimers (Supplementary Fig. 4D, E). In the LRRK1 dimer, the kinase domain is occluded by the ankyrin repeats of the neighboring LRRK1 molecule in the dimer, in LRRK2, the kinase domain is occluded by the LRRs of the same monomer. This indicates that the current LRRK dimer structures represent intrinsically inactive forms of the molecule.

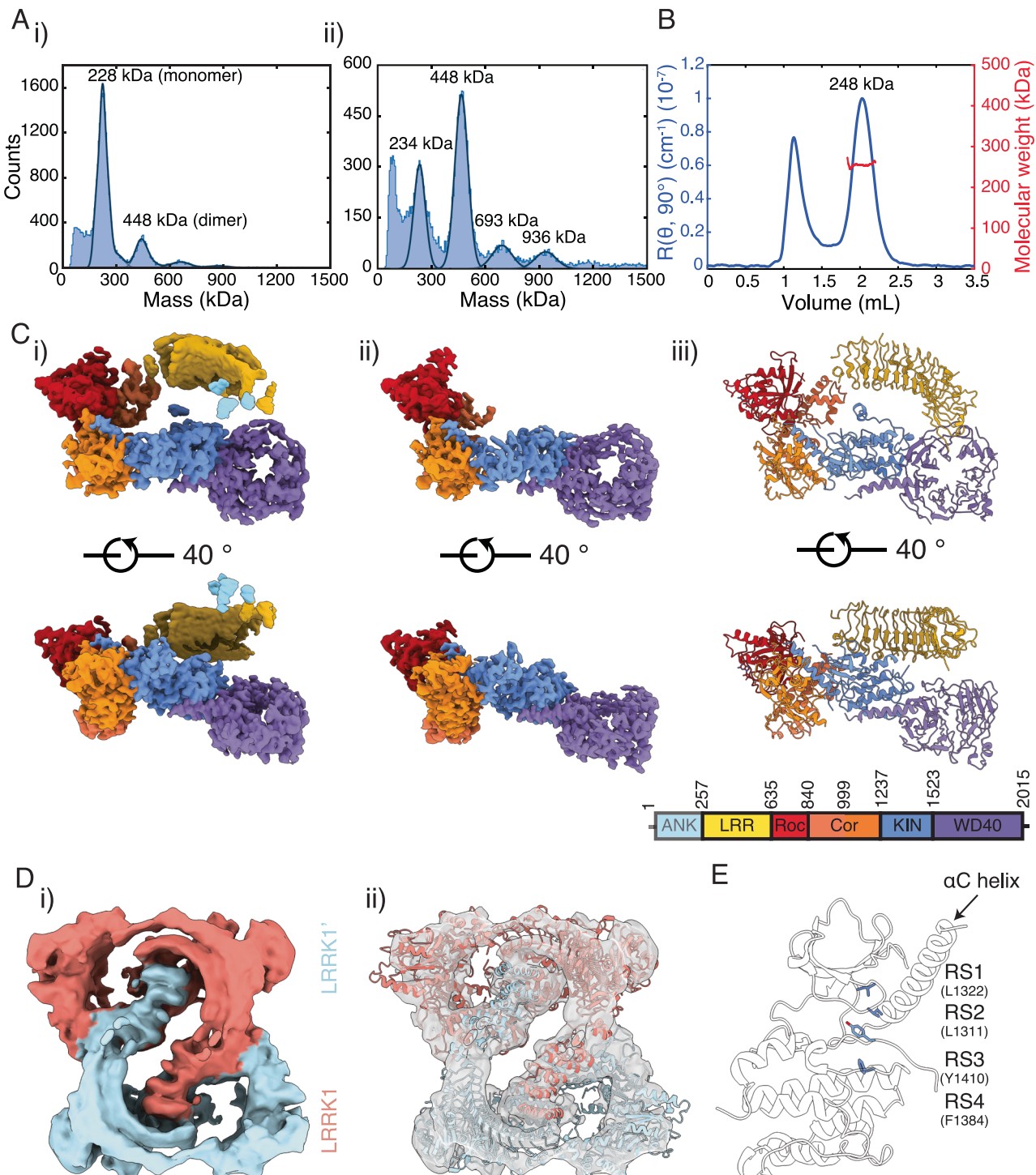

**Fig. 1 | Structure of LRRK1 and the inactive kinase domain of LRRK1. A** Mass photometry mass distributions for, i) the LRRK1 monomer, and ii) the LRRK1 dimer fractions. **B** SEC-MALS chromatogram for the LRRK1 monomer. **C** Cryo-EM map and model of the LRRK1 monomer, i) map from global non-uniform refinement, ii) map from local refinement of the C-terminal RCKW domains, iii) atomic model of the LRRK1 monomer, colored according to the schematic. **D** Cryo-EM analysis of the LRRK1 dimer, i) cryo-EM map of the LRRK1 dimer, ii) two copies of the LRRK1 monomer fit in the LRRK1 dimer map. **E** The broken kinase R-Spine (RS) in the kinase domain in the LRRK1 monomer, indicating that the kinase domain is in the inactive conformation. Maps in **C** were generated after post-processing in *deepEMhancer*[59], maps in **D** were generated after sharpening in *Cryosparc*.

## Structural scaffolds in the kinase and GTPase Roc-COR domains control LRRK1 activity

Two structural scaffolds in the COR-B and kinase domains of LRRK1 serve to link the two domains and tightly regulate the activity of the kinase domain, controlling the kinase inactive-to-active transition. First, in LRRK1, the αC helix (residues 1288–1313) is unusually long,

approximately twice as long compared to the αC helix in LRRK2 and other comparable kinases. This elongated helix acts as a structural scaffold, linking the kinase N-lobe and the COR-B domain, and forms unique contacts with the COR-B domain specific to LRRK1 (Fig. 3A–C, Supplementary Fig. 6). Typically, the αC helix transitions from an 'out' to 'in' position upon kinase activation, allowing an

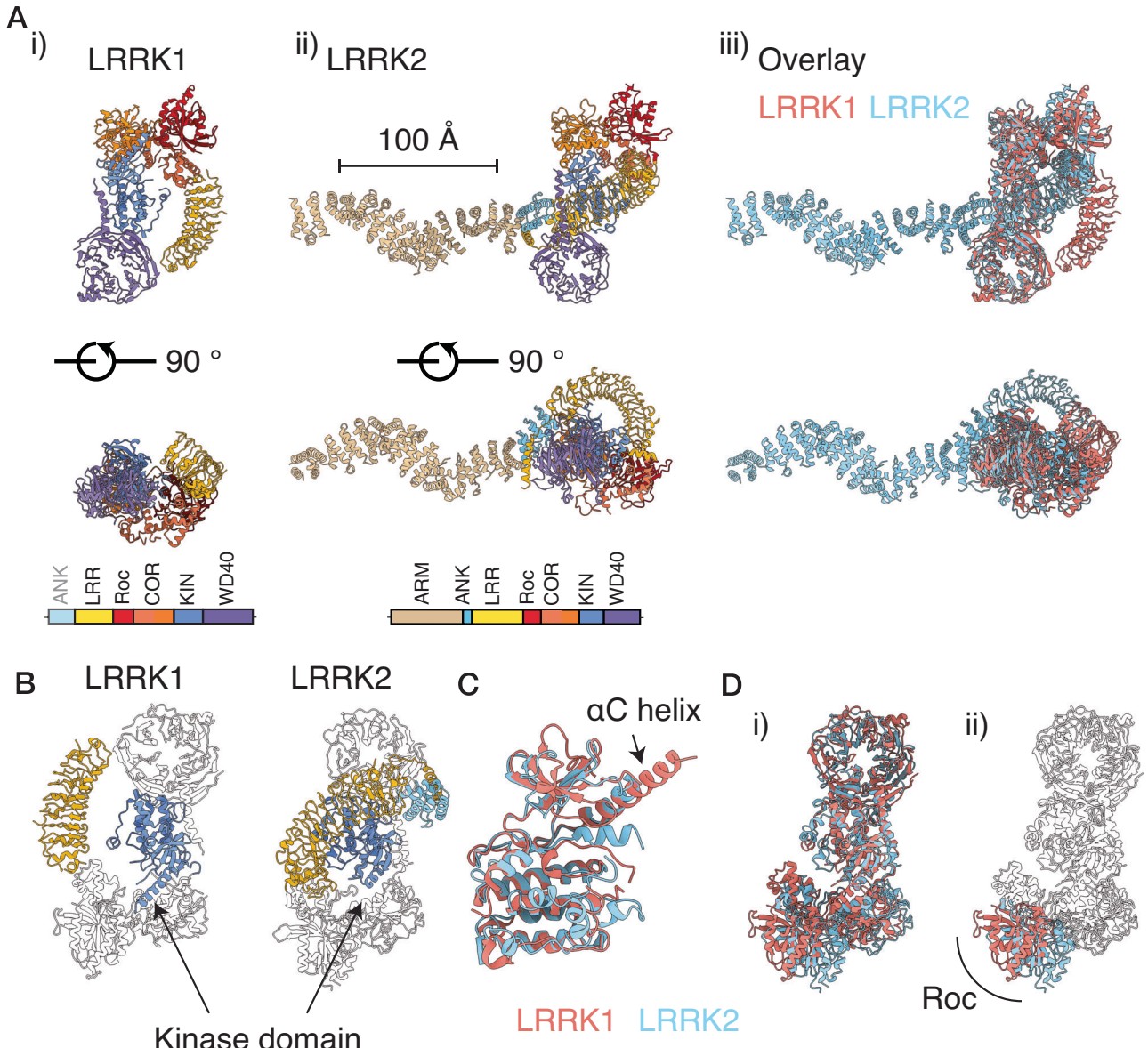

**Fig. 2 | Structural comparison of the LRRK1 and LRRK2 monomers. A** i) The structure of the LRRK1 monomer, ii) the LRRK2 monomer[29] (PDB: 7LHW, EMD-23352, residues 1–543 from predicted Alphafold[39,60] model AF-Q5S007, and iii) an overlay. Structures are displayed on the same scale. **B** Relative position of the LRRs and the kinase domain for LRRK1 and LRRK2 showing that the kinase domain in LRRK2 is occluded by the LRRs. **C** Overlay of the kinase domain of LRRK1 and LRRK2. **D** Overlay of the C-terminal RCKW domains of LRRK1 and LRRK2, aligned on the kinase domain, i) shows the overlay, ii) highlights the shifted position of the Roc domain, which accommodates the altered positions of the LRRs.

overall rearrangement of the kinase core to allow substrate phosphorylation[35]. Interactions involving the αC helix often serve to be a key regulator of kinase activity in multidomain kinases or kinase complexes[36,37].

The second key scaffold in the LRRK1 catalytic domains is the DK helix (the COR-B α2 helix, also referred simply as the COR-B helix, residues 1132–1144). The DK helix is positioned at an approximately 30° angle relative to the αC helix. In contrast to the inactive LRRK2 structure, where the DK helix forms several electrostatic contacts with the αC helix[29,38], the LRRK1 DK helix is shifted by ~3 Å compared to LRRK2 and thus does not form extensive electrostatic contacts with the αC helix. Hydrophobic residues on the face of the DK helix opposite the αC helix pack tightly with COR-B domain, stabilizing this position of the helix (Fig. 3D). The C-terminal end of the DK helix sits against the Roc domain, creating an interface which is critical for LRRK2 regulation[38].

We tested the effect of a set of mutations at the αC helix/COR-B interface and the Roc/COR-B interfaces using an in vitro kinase assay, measuring Rab7A phosphorylation by purified, recombinant LRRK1 as a readout (see Methods, Supplementary Figs. 7–9). We expressed a set of mutations that spanned the conserved region of the αC helix as well as the unique extension specific to LRRK1 (Fig. 3B, C, Fig. 4A, Supplementary Fig. 7). Additionally, we assessed the impact of all these mutations on the thermal stability of LRRK1 using differential scanning fluorimetry to understand their effects on protein stability (Supplementary Fig. 10).

First, we investigated the conserved region of the αC helix/COR-B interface shared by LRRK1 and LRRK2 (Fig. 4A-B). We investigated the effects of the R1305A, Q1306A/S1309A, and F1301A mutations. All mutations were expressed well in vitro. The R1305A and F1301A mutations showed destabilization relative to the wild-type (WT) ($\Delta T_M$ −2.0 °C for R1305A, −1.7 °C for F1301A), while the Q1306A/S1309A

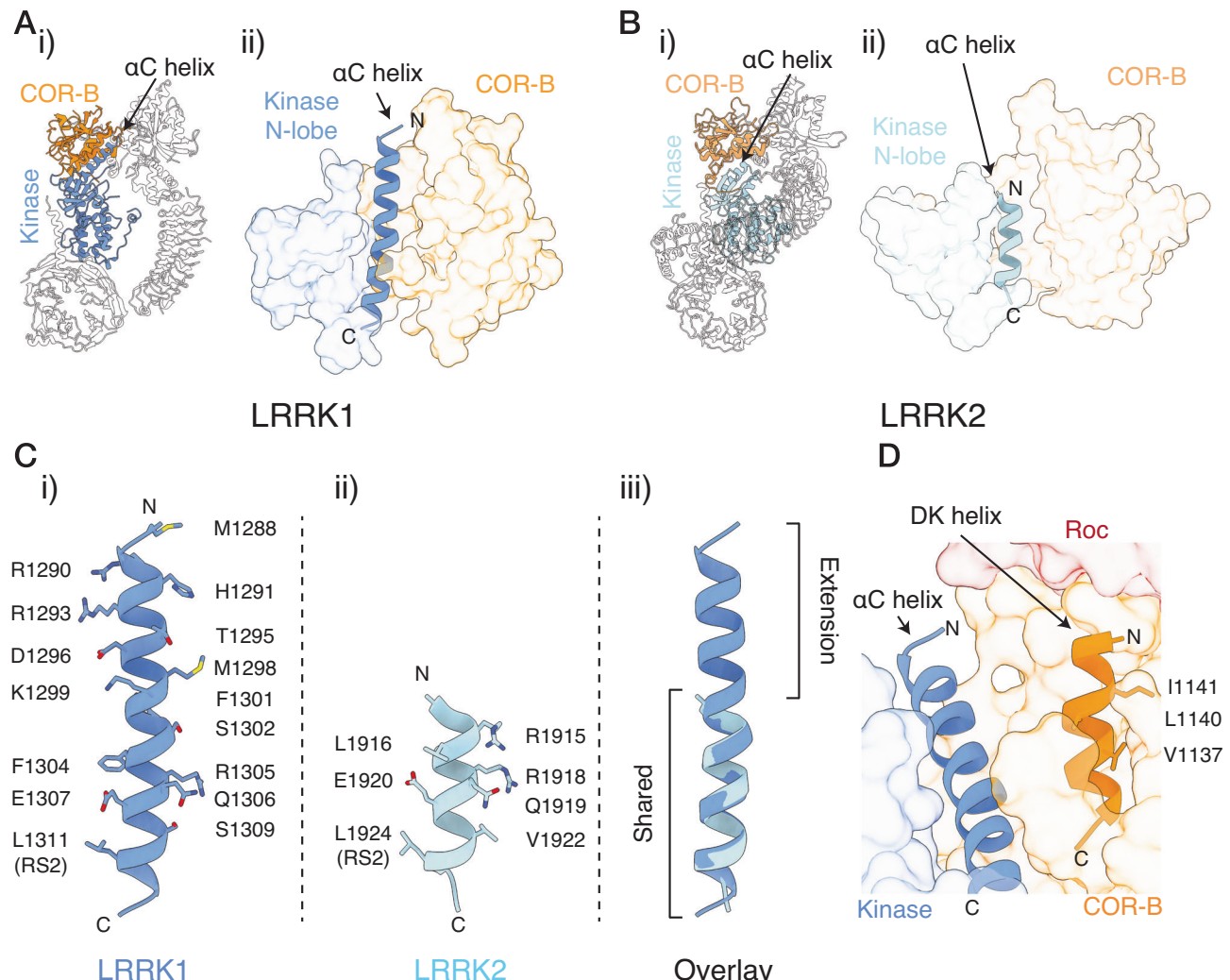

**Fig. 3 | Interdomain interfaces regulating the LRRK1 kinase domain. A** The interface formed by the LRRK1 kinase and the COR-B domain. **B** The interface formed between the kinase N-lobe, αC helix and the COR-B domain in LRRK1, i) and LRRK2 ii). **C** The αC helix in LRRK1, i), LRRK2 ii) and an overlay, iii), with residues along the LRRK1 and LRRK2 αC helix indicated in i) and ii), the shared region, and the unique LRRK1 extension indicated in iii) (see also Supplementary Fig. 6). **D** The position of the COR-B DK helix relative to the kinase αC helix and the Roc domain (shaded red). Hydrophobic residues in the DK helix which stabilize the position of the helix are indicated. In **A**, **B**, and **D**, the surfaces displayed are solvent-exposed surfaces.

mutant showed increased stability ($\Delta T_M$ 0.8 °C, Supplementary Fig. 10). The F1301A mutation does not alter Rab7A phosphorylation by LRRK1 (Fig. 4A, Supplementary Fig. 7A, $p = 0.33$ relative to WT). The R1305A and Q1306A/S1309A mutations both reduced Rab7A phosphorylation relative to WT LRRK1. Specifically, the R1305A mutation diminished Rab7A phosphorylation to the same level as the canonical K1270M kinase-dead mutation (Fig. 4A, Supplementary Fig. 7A, $p = 0.002$ relative to WT, $p = 0.62$ relative to K1270M). Conversely, Q1306A/S1309A reduced Rab7A phosphorylation, but not to the same level as the K1270M mutation (Fig. 4A, Supplementary Fig. 7B, $p = 0.0041$ relative to WT, $p = 0.04$ relative to K1270M). In the inactive-kinase state LRRK1 structure, the R1305 sidechain is well-defined in the density, protruding into a small pocket formed by the COR-B domain (Fig. 4C). The sidechains of Q1306 and S1309 are not well-defined in the density, however they sit adjacent to a regulatory loop known to be phosphorylated by PKC (Fig. 4C, discussed below). While an experimentally-determined active-kinase structure of LRRK1 is not available, the predicted Alphafold[39] model of LRRK1 has the kinase in an active conformation, and is reminiscent of the LRRK2 active conformation[31]. In the Alphafold model, the COR-B DK helix undergoes a large rearrangement upon activation, similar to the rearrangement observed in the active LRRK2 structure[31]. This rearrangement leads to the formation of a hydrogen bond between D1135 in the DK helix and R1305 in the αC helix (Fig. 4C, D), presumably stabilizing the active state of the kinase. Likewise, in the predicted active state structure, Q1306 informs contacts with the kinase activation loop, implying its involvement in stabilizing the kinase active state. Therefore, removal of both R1306 and Q1306/S1309 greatly decrease or abolish LRRK1 kinase activity, by removing residues essential for the LRRK1 inactive-to-active state transition.

Adjacent to the conserved αC helix region, a loop containing two residues (S1074, T1075) have been shown to be phosphorylated by PKC[40] sit between the base of the DK helix and the αC helix, and close to the activation loop (Fig. 4E). Phosphorylation of these residues by PKC leads to the activation of the LRRK1 kinase[40]. Notably, the R-Spine residue L1311 and the kinase activation loop is near the loop phosphorylated by PKC (Fig. 4E), suggesting that the residues phosphorylated by PKC are located at a key regulatory interface in the protein. In the unphosphorylated state, Q1306 and S1309 in the αC helix face the loop and interact with S1074 and T1075, stabilizing the 'inactive' position of this loop. As discussed earlier, mutation of Q1306 and S1309 to alanine results in the loss of

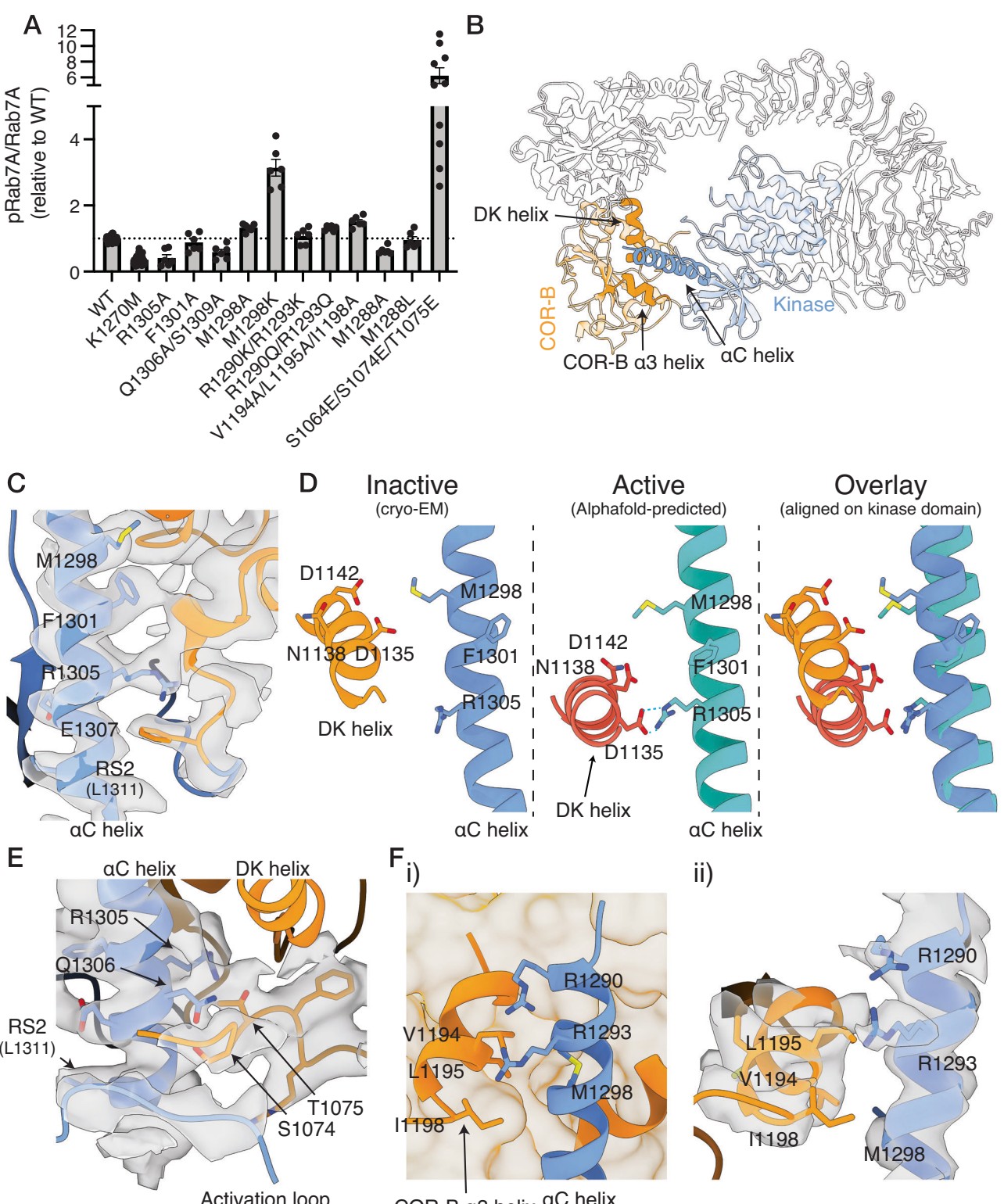

**Fig. 4 | The kinase αC helix/COR-B interface. A** Effect of mutations in the kinase αC helix/COR-B interface on Rab7A phosphorylation by recombinant LRRK1. Results presented are the quantification of multiple kinase assays from at least three independent protein preparations (see "Methods", Supplementary Fig. 7A–C for representative membrane images and Supplementary Fig. 8A–C for complete membrane images, values are mean ± SEM, $n = 6–36$ independent assays from 9 independent protein preparations (WT, K1270M), 5 independent protein preparations (S1064E/S1074E/T1075) or 3 independent protein preparations (all other mutants)). **B** Related to **C**–**E**, structure of LRRK1, with the DK helix, kinase αC helix and the hydrophobic COR-B α3 helix indicated in (**C**). **C** Cryo-EM density

supporting the position of F1301 and R1305 at the conserved, C-terminal end of the αC helix. **D** Structural changes between the cryo-EM model, which is in the inactive state, and the active-state Alphafold model, highlighting a predicted interaction between R1305 and D1135 in the COR-B DK helix. **E** Cryo-EM density supporting the position of S1074 and T1075 at the base of the αC helix. **F** Interaction between the extended αC helix and the α3 helix in the COR-B domain, i) showing the position of the helix, ii) cryo-EM density supporting this interface. In **C**, **E** and **F**ii, the surface shown is the composite cryo-EM density map. In **F**i, the surface shown is a solvent-exposed surface. Source data is provided as a source data file.

Rab7A phosphorylation by LRRK1, demonstrating their role in LRRK1 activation. This loop beyond N1071, which is the last residue included in the model, is poorly resolved in the density. Therefore, the third PKC phosphorylation site (S1064) is thus not included in the model. Contouring the density map at a low threshold indicates the loop likely continues past the αC helix, towards the kinase N-lobe, (Supplementary Fig. 11). The triple-phosphomimetic mutation, S1064E/S1074E/T1075E mimics PKC phosphorylation, in cellulo increasing LRRK1 Rab7A phosphorylation, albeit not to the levels by PKC phosphorylation of these residues[40]. Similarly, we observed that the recombinant triple-phosphomimetic increases Rab7A phosphorylation 5-fold over recombinant WT LRRK1 (Fig. 4A, Supplementary Fig. 7A, $p = 0.0024$). The triple-phosphomimetic mutation destabilizes LRRK1 ($\Delta T_M$ −1.2 °C, Supplementary Fig. 10), consistent with a destabilization of this regulatory interface. Phosphorylation of these residues would result in a rearrangement of this critical regulatory interface involving the αC helix, the activation loop and the COR-B domain allowing kinase activation.

Next, we investigated the extension of the αC helix, which forms unique contacts with the COR-B domain in LRRK1, generating a series of mutations in this interface (Fig. 3C). First, we investigated residue M1298, which is the starting residue in the LRRK1 αC helix extension, sitting adjacent to several negatively charged residues in the DK helix, but not directly contacting the DK helix (Fig. 4C, D). The M1298A mutation modestly increased Rab7A phosphorylation by approximately 1.3-fold (Fig. 4A, Supplementary Fig. 7C, $p = 0.0006$ relative to WT). Strikingly, the M1298K mutation greatly increased Rab7A phosphorylation, by approximately 3-fold (Fig. 4A, Supplementary Fig. 7C, $p = 0.0004$ relative to WT). Notably, the M1298K mutation is positioned to introduce a new contact between the αC helix and negatively charged residues on the DK helix. This finding suggests that the M1298K mutation induces the formation of a stabilizing contact between the αC helix and the COR-B DK helix, promoting the active state of the helix and establishing interdomain contacts that stabilize the kinase and COR-B domains in a conformation favorable for kinase activity. Both mutations modestly destabilize LRRK1 ($\Delta T_M$ −0.4 °C for M1298A, −0.3 °C for M1298K, Supplementary Fig. 10).

The αC helix forms an additional contact with the COR-B domain at the extreme N-terminus of the αC helix, where it contacts a short hydrophobic helix in the COR-B domain (the COR-B α3 helix, formed by residues 1190–1200, Fig. 4F). This interaction between the kinase and COR-B domain is unique to LRRK1, as it is created by the extended αC helix. The contact involves R1290/R1293 on the αC helix and a hydrophobic patch composed of residues V1994/L1195/I1198 on the COR-B α3 helix. The R1290A/R1293A double-mutation were expressed at very low levels and could not be biochemically characterized. However, the more conservative R1290K/R1293K and R1290Q/R1293Q mutations were expressed to levels analogous to WT LRRK1. The R1290K/R1293K mutation did not alter LRRK1 Rab7A phosphorylation (Fig. 4A, Supplementary Fig. 7B, $p = 0.49$ relative to WT), while the R1290Q/R1293Q mutation modestly increased Rab7A phosphorylation by 1.3-fold (Fig. 4A, Supplementary Fig. 7B, $p = 0.0001$ relative to WT). Both mutations resulted in stabilization of LRRK1 ($\Delta T_M$ 1.0 °C for R1290K/R1293K, 0.1 °C for R1290Q/R1293Q, Supplementary Fig. 10). Next, we tested the effect of removing the hydrophobic residues in the hydrophobic COR-B α3 helix. The V1994 A/L1195A/I1198A mutation resulted in a modest but consistent increase of Rab7A phosphorylation by approximately 1.5-fold (Fig. 4A, Supplementary Fig. 7A, $p = 0.0003$ relative to WT). This mutation destabilized LRRK1 ($\Delta T_M$ −1.4 °C, Supplementary Fig. 10). Furthermore, we examined the substitution of M1288, which caps the αC helix. The M1288A mutation resulted in lower protein expression relative to WT LRRK1 and, consistent with this, a decrease in thermal stability ($\Delta T_M$ −1.5 °C, Supplementary Fig. 10). The M1288A mutation modestly decreased Rab7A

phosphorylation by LRRK1, but not to the levels of the K1270M kinase-dead mutant (Fig. 4A, Supplementary Fig. 7A, $p = 0.0003$ relative to WT, 0.0007 relative to K1270M). Some caution should be taken in interpreting this result as the low expression level of M1288A made accurately measuring the kinase concentration difficult. The M1288L mutation had expression levels comparable to the WT and did not significantly alter Rab7A phosphorylation (Fig. 4A, Supplementary Fig. 7C, $p = 0.68$ relative to WT). Moreover, the M1288L mutation stabilized LRRK1 ($\Delta T_M$ 1.0 °C, Supplementary Fig. 10). Overall, this shows that residues at the N-terminal end of the αC helix are critical for protein stability, as their removal disrupts protein expression. Furthermore, the unique contacts formed between this region of the αC helix and the COR-B domain contribute to stabilizing the LRRK1 inactive state, as disrupting this interface results in modest activation of the kinase.

We next investigated the effect of a set of mutations in the Roc/COR-B interface formed by the DK helix (Fig. 5A-B). The interface is predominantly formed by the COR-B domain DK helix, the adjacent α1 helix in the COR-B domain, and the α3 and α4 helices in the Roc domain. The kinase domain does not directly contact the Roc domain, and the COR-B domain effectively serving to scaffold the two domains. Activating mutations in this interface are present in both LRRK1 and LRRK2[7,41,42]. Mutation of K746 in the Roc domain, which sits at the interface (Fig. 5C), is a known activating LRRK1 mutation[7], as is the analogous mutation in LRRK2 (R1441G[42]). Consistent with previous findings, the K746G mutation substantially increased Rab7A phosphorylation by 12-fold in our in vitro assay (Fig. 5A, Supplementary Fig. 7D, $p = 0.0148$ relative to WT). In the cryo-EM model, K746 sits above the N-terminal end of the DK helix (Fig. 5B, D). In the Alphafold model of the active-state of LRRK1, the DK helix undergoes a 15 ° shift relative to the inactive-state cryo-EM structure, resulting in the formation of extensive contacts between the DK helix and the αC helix, and bring it closer to the kinase activation loop (Fig. 5B–E), which places K746 further away from the DK helix (Fig. 5D). The K746G mutation destabilized LRRK1 ($\Delta T_M$ −2.7 °C, Supplementary Fig. 10), indicating that the mutation destabilizes this interface, presumably leading to kinase activation. We also investigated additional mutations in this interface, including W1144A, which sits on the C-terminal end of the DK helix, a W1144A/K746G double-mutant, R1030A and R1034A, both of which sit on the COR-B α1 helix. Except for the R1030A mutant, which did not express to levels sufficient to allow biochemical characterization, all mutations reduced or abolished Rab7A phosphorylation (Fig. 5A, Supplementary Fig. 7D). Furthermore, all mutations destabilized LRRK1 ($\Delta T_M$ −1.2 °C for W1144A, $\Delta T_M$ −2.2 °C for W1144A/K746G, $\Delta T_M$ −1.5 °C for R1034A, Supplementary Fig. 10). In the inactive state, W1144 is located close to K746 (Fig. 5D). In the Alphafold model of the active state, the shift in the DK helix places W1144 close to the kinase activation loop, analogous to a conformational change seen in the active state of LRRK2[31]. The W1144A mutation reduced Rab7A phosphorylation to the same level as the K1270M kinase-dead mutant (Fig. 5A, Supplementary Fig. 7D, $p = 0.0005$ relative to WT, 0.55 relative to K1270M), the W1144A/K746G double-mutant reduced Rab7A phosphorylation, but not to the level of the K1270M kinase-dead mutant (Fig. 5A, Supplementary Fig. 7D, $p < 0.0001$ relative to WT, 0.0101 relative to K1270M). Similarly, the R1034A mutation likewise reduced Rab7A phosphorylation to the level of the K1270M kinase-dead mutant (Fig. 5A, Supplementary Fig. 7D, $p < 0.0001$ relative to WT, $p = 0.23$ relative to K1270M). In the cryo-EM model of the inactive structure, R1034 sits alongside the DK helix, in both the cryo-EM model of the inactive state, and the Alphafold model of the active state, R1034 interacts with the DK helix, stabilizing the active conformation, rationalizing the kinase-inactivating properties of the R1034A mutant (Fig. 5D). Thus, the Roc/COR-B interface through the DK helix is key for regulating LRRK1 kinase activity, despite its distant location from the key kinase regulatory regions in the inactive state of LRRK1. Disrupting

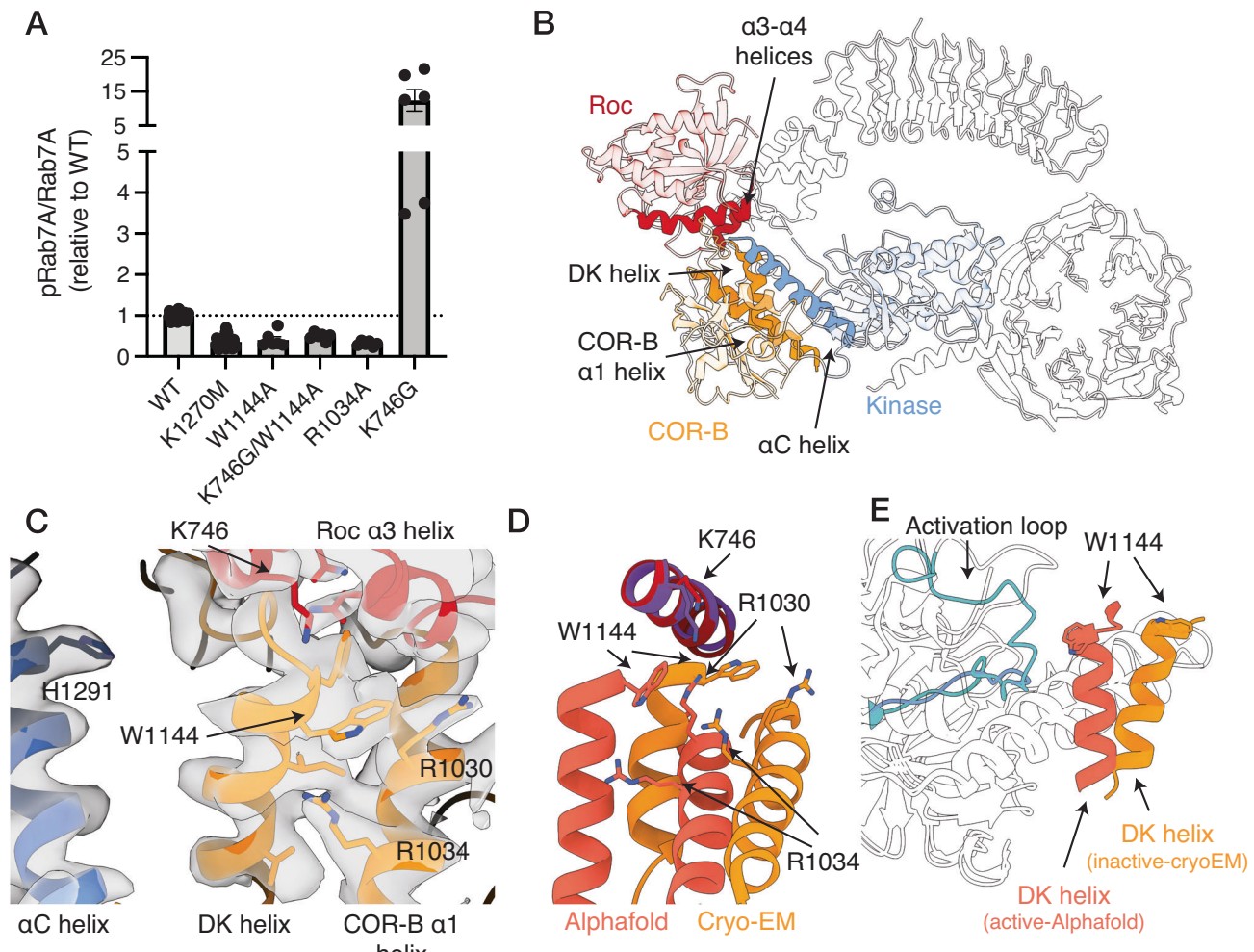

**Fig. 5 | The Roc-COR-kinase interface. A** Effect of mutations in this interface on Rab7A phosphorylation by recombinant LRRK1. Results presented are the quantification of multiple kinase assays from at least three independent protein preparations (see "Methods", Supplementary Fig. 7D for representative membrane images and Supplementary Fig. 8D for complete membrane images, values are mean ± SEM, $n = 6$–36 independent assays from 9 independent protein preparations (WT, K1270M) or 3 independent protein preparations (all other mutants)). **B** Related to **C**–**E**, structure of LRRK1, with the position of the αC helix, COR-B DK and α1 helices and K746 indicated in the structure. **C** Cryo-EM density supporting the Roc/COR interface, additionally showing that the αC helix is positioned away from the interface in the inactive state. **D** Structural changes between the inactive-state cryo-EM model, and the active-state Alphafold model, highlighting a change in the position of K746 relative to the DK helix, and in the position of R1034. **E** Overlay of the DK helix/activation loop interface between the cryo-EM and Alphafold models, highlighting the movement of the DK helix towards the activation loop. In **C** the map shown is the composite cryo-EM density map. Source data is provided as a source data file.

the interface either results in kinase hyper-activation, possibly through lifting inhibitory Roc/COR-B/kinase interactions, or kinase inactivation, by preventing the formation of interactions required to stabilize the active state of the kinase, analogous to the R1305 mutation in the kinase αC helix.

LRRK1 is regulated by the EGFR, with EGFR tyrosine phosphorylation of Y971 resulting in kinase inhibition[13]. The effects of Y971 phosphorylation are surprising, as Y971 is in the COR-A domain, on the surface of the molecule and distant from canonical regulatory elements in the Roc and kinase domains (Fig. 6B). The Y971F mutation activates LRRK1 in vitro and has been proposed to function through abolishing EGF-mediated inactivation, although the Y971F mutation is activating in the absence of EGF simulation[7,9,13]. Consistent with this, we observed that recombinant Y971F resulted in 3-fold greater Rab7A phosphorylation relative to WT LRRK1 (Fig. 6A, Supplementary Fig. 7E, $p < 0.0001$). The Y971F mutation modestly increased LRRK1 stability ($\Delta T_M$ 0.5 °C, Supplementary Fig. 10). The robust activation of the Y971F mutation, both in the presence and absence of EGF stimulation was striking, as the Y971F mutation only removes the hydroxyl group of

Y971, removing a tyrosine phosphosite, and disrupting any hydrogen-bonds formed by Y971. To further study the mechanism of the Y971F mutation, we studied several additional substitution mutations, Y971A, Y971Q and Y971L. The Y971A mutation did not alter Rab7A phosphorylation ($p = 0.36$ relative to WT), the Y971Q mutation reduced Rab7A phosphorylation to the same level as the K1270M kinase-dead mutant (Fig. 6A, Supplementary Fig. 7E, $p = 0.0004$ relative to WT, $p = 0.12$ relative to K1270M), and the Y971L mutation resulted in increased Rab7A phosphorylation by approximately 2-fold relative to WT LRRK1, but not to the same level as the Y971F mutant (Fig. 6A, Supplementary Fig. 7E, $p = 0.0005$ relative to WT, $p = 0.0001$ relative to Y971F). The Y971Q and Y971L mutations did not greatly alter LRRK1 thermal stability, in contrast, the Y971A mutation destabilized LRRK1 ($\Delta T_M$ 0.4 °C for Y971Q, $\Delta T_M$ 0.2 °C for Y971L, $\Delta T_M$ −1.6 °C for Y971A, Supplementary Fig. 10).These results showed that removal of the hydrophobic tyrosine sidechain completely does not activate LRRK1 (as with the Y971A mutation), while maintaining the hydrophobic sidechain but removing the tyrosine hydroxyl group activates LRRK1 (as with the Y971L/F mutations). Finally, the Y971Q mutation, which

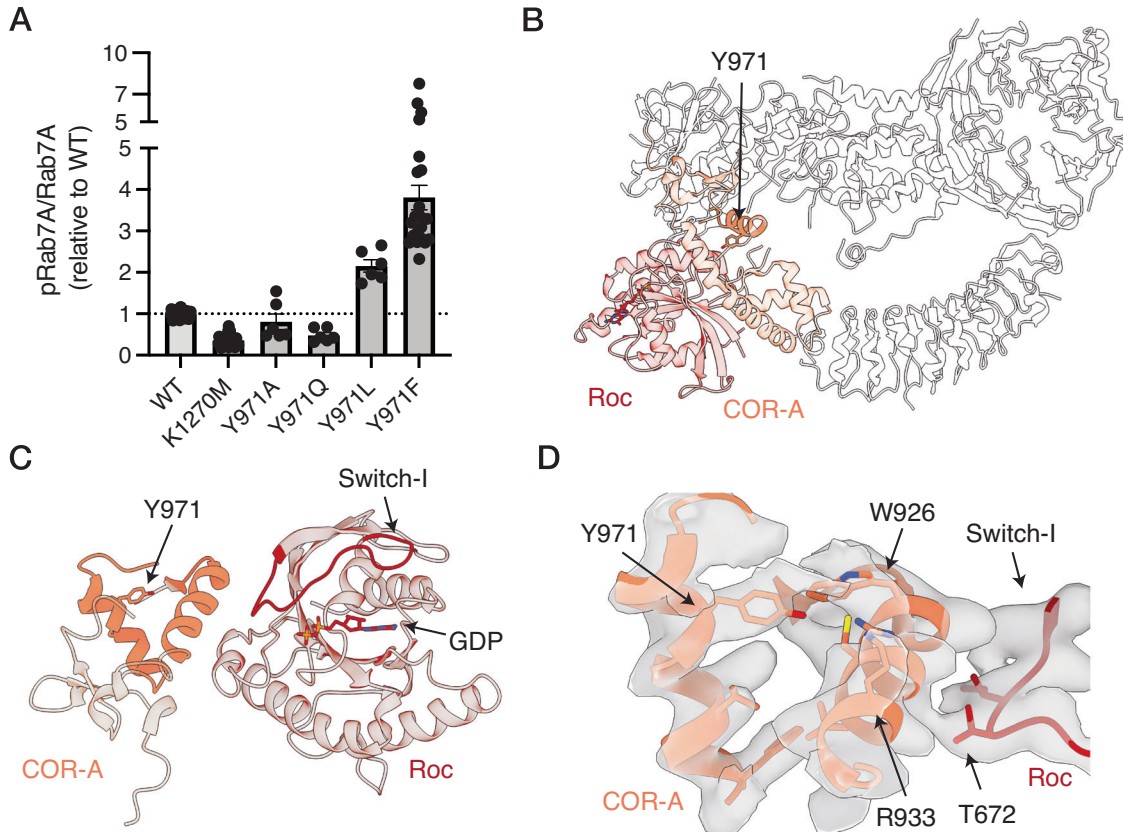

**Fig. 6 | Activation of LRRK1 by mutation of Y971, and location in the structure.**
**A** Effect of Y971 substitution mutations. Results presented are the quantification of multiple kinase assays from at least three independent protein preparations (see "Methods", Supplementary Fig. 9E for representative membrane images and Supplementary Fig. 10 for complete membrane images, values are mean ± SEM, $n = 6$–36 independent assays from 9 independent protein preparations (WT,

K1270M, Y971F) or 3 independent protein preparations (all other mutants)). **B** Structure of LRRK1, with the position of Y971, and the Roc and the COR-A domains indicated. **C** Location of Y971 in the COR-A domain relative to the Switch-I motif and the bound GDP in the Roc domain. **D** Cryo-EM density supporting the position of Y971. In **D** the map shown is the composite cryo-EM density map. Source data is provided as a source data file.

removes the tyrosine phosphosite and maintains any hydrogen bonds formed by the hydroxyl group of Y971, inactivates LRRK1. These results suggest that phosphorylation of Y971 disrupts interactions involving the COR-A domain and the enzymatic domain of LRRK1. Y971 is near the Switch-I motif in the Roc domain, which responds to guanidine nucleotide binding, and the bound GDP (Fig. 6C, D) so phosphorylation of Y971 may serve to indirectly modulate LRRK1 kinase activity through the Roc domain. This allows the COR-A domain to serve as a hub for regulation of the LRRK1 kinase by EGFR, and possibly other regulatory partners.

LRRK1 is also regulated through phosphorylation by CDK11 at T1427, and PLK1 at S1817[43,44]. Both residues are not resolved in the structure. T1427 lies in the kinase domain activation loop, so phosphorylation of this residue likely stabilizes the LRRK1 kinase domain active state, a well-described mechanism of kinase activiation[45]. S1817 is not resolved in the structure, lying in a disordered loop in the WD40 domain. Phosphorylation of this residue may serve to alter LRRK1 activity by altering its interaction with other interacting partners.

To complement our in vitro studies, we transfected the U2OS human bone osteosarcoma epithelial cell line with plasmids expressing WT LRRK1 and the R1290A/R1293A, K1270M, Y971F, and K746G mutants (Fig. 7). Consistent with our in vitro observations that the K746G mutant and the R1290A/R1293A mutant express poorly in insect cells, the expression level in the U2OS cells was approximately half compared to WT (Fig. 7A, Bi). The Y971F mutant expressed to similar levels as WT LRRK1 (Fig. 7A, Bi). The

level of R1290A/R1293A mutant is slightly lower than the WT but does not reach statistical significance ($p = 0.3265$, Fig. 7Bii). Notably, and consistent with our in vitro kinase experiments, and the literature[7,13], the Y971F and K746G mutants resulted in robust kinase activation compared to WT LRRK1. The K746G and Y971F mutants both showed a 3-fold increase in phosphorylation relative to WT ($p < 0.0001$ for both K746G and Y971F, relative to WT, Fig. 7A, Bii). It has been previously reported that LRRK1 displays endosomal localization, as shown by colocalization with several endosomal markers including the late endosomal marker Rab7[9,12,13]. We hypothesized that mutations that altered LRRK1 kinase activity would affect the presence of LRRK1 in Rab7-positive compartments. Using super-resolution microscopy, we measured the degree of LRRK1 colocalization to exogenously expressed Rab7. As expected, all LRRK1 variants displayed Rab7 colocalization. The hyperactive K746G and Y971F mutants displayed increased colocalization relative to WT LRRK1 ($p < 0.0001$ for K746G, $p < 0.0001$ for Y971F, Fig. 7C, D). The K1270M kinase-dead mutant and the R1290K/R1293K mutant did not significantly alter LRRK1 localization over WT LRRK1 ($p = 0.2173$ for R1290A/R1293A, $p = 0.1558$ for K1270M, Fig. 7D), despite showing a decreased ability to phosphorylate Rab7. To note, the hyperactive mutants K746K and Y971F increase late endosomal clustering to the perinuclear area. These findings are in line with previous reports suggesting that the phosphorylation of S72-Rab7 by LRRK1 enhances the binding of Rab7 to the dynein adapter protein RILP, thereby facilitating endosomal retrograde transport[9].

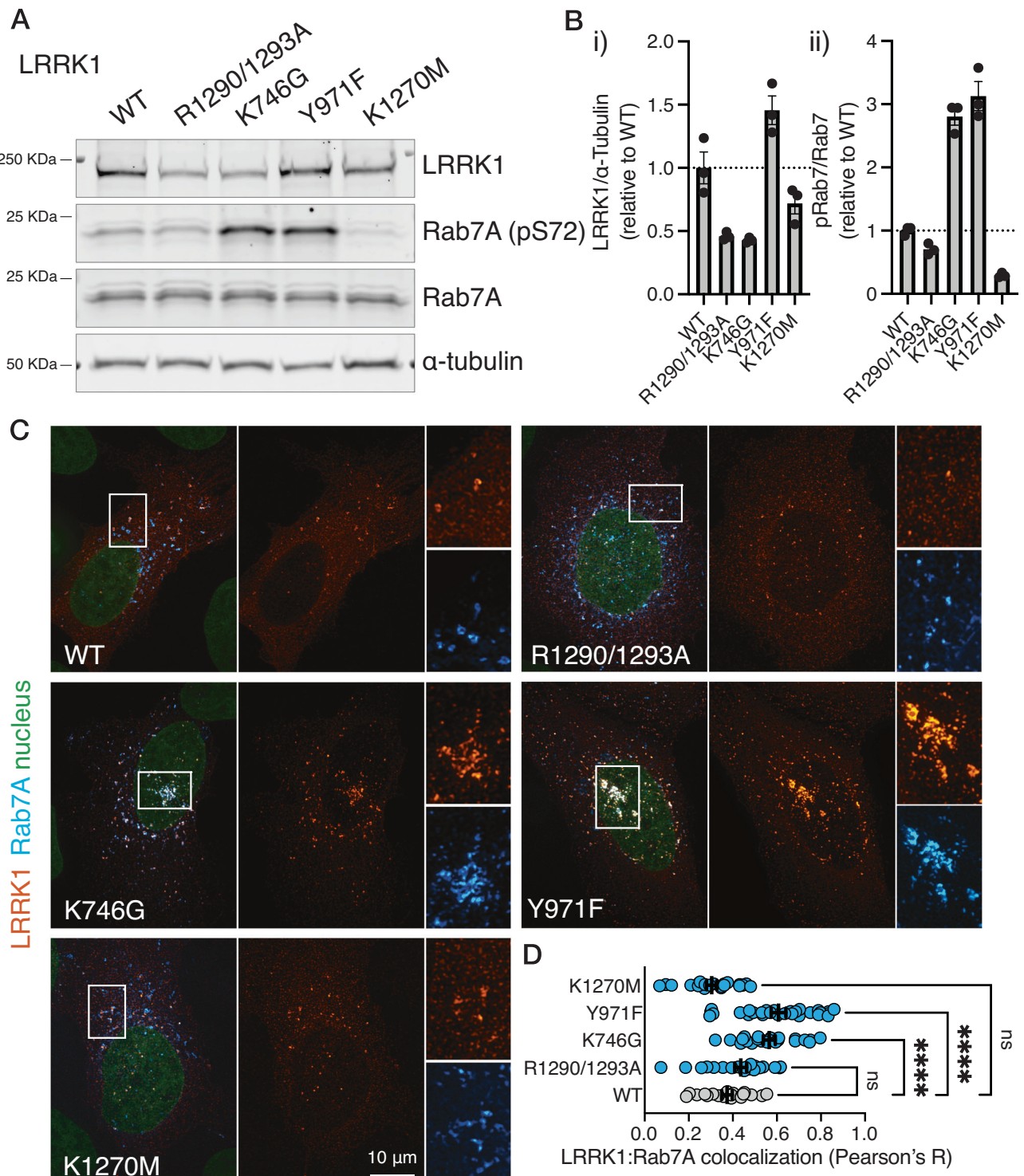

**Fig. 7 | Activity and cellular localization of LRRK1 and LRRK1 variants in vitro.**
**A** Western blot showing phosphorylation of Rab7A, and relative LRRK1 expression level of the indicated LRRK1 mutants. **B** Quantification of membranes shown in (**A**), i) quantification of total LRRK1, normalized to α-tubulin, ii) quantification of pRab7, normalized to total Rab7. Data are mean ± SEM ($n = 3$ independent experiments). **C** Representative confocal images of U2OS cells transfected with the indicated LRRK1 variant, stained for total LRRK1 (flag) and GFP-Rab7A. **D** Quantification of LRRK1/Rab7A colocalization. Data are mean ± SEM ($n = 22$–28 cells per group over one independent experiment). One-way ANOVA with Dunnett's was used for multiple comparisons analysis. Effect size was measure as $R^2 = 0.4322$. **** means $p$ value < 0.0001. Source data is provided as a source data file.

## The kinase domain interacts with the WD40 domain at the C-terminal helix

Similar to LRRK2, the WD40 domain sits against the C-lobe of the kinase, with the WD40 C-terminal helix effectively contributing an additional helix to the kinase C-lobe fold[28] (Supplementary Fig. 12A).

The N-terminal end of the helix forms extensive interactions with the kinase C-lobe, largely through the aromatic residues W1989, F1997, Y1998 and Y2001 (Supplementary Fig. 12A, B). The helix is slightly bent, so the C-terminal end of the helix does not contact the C-lobe of the kinase. The exact role for the C-terminal helix is unclear. In LRRK2, the

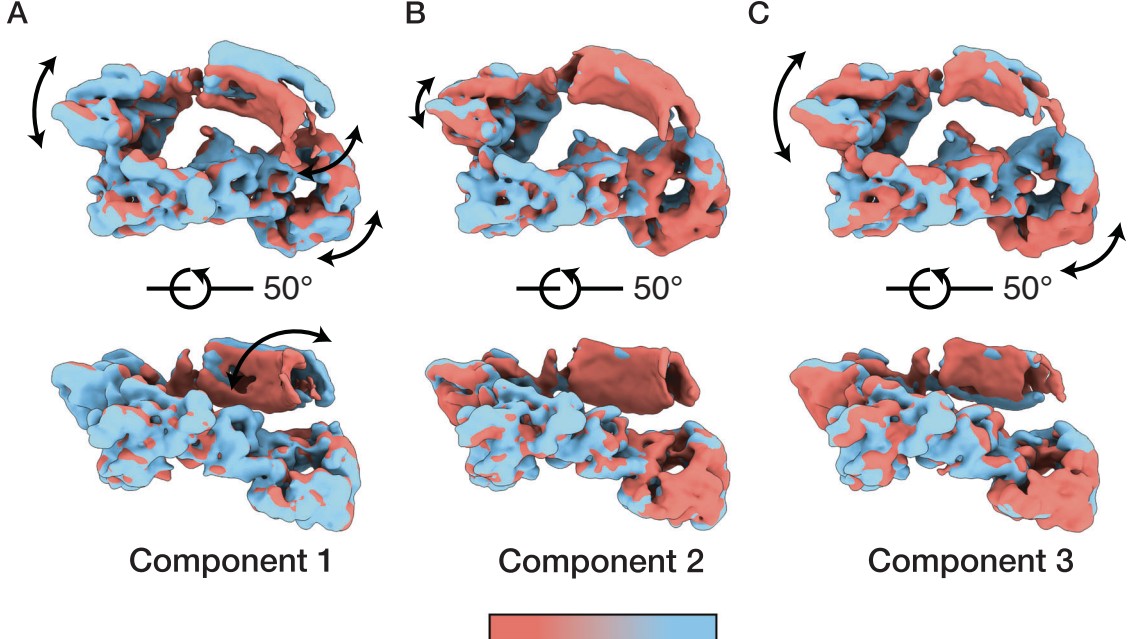

**Fig. 8 | 3D variability analysis of the LRRK1 monomer. A** variability component 1, showing a movement of the Roc domain, and a large movement of the leucine-rich repeats. **B** variability component 2, showing little movement. **C** variability component 3, showing a movement of the Roc domain. The arrows indicate the movement experienced by LRRK1 along the variability component, with the density maps colored according to their position along the variability component. See also, Supplementary Movie 1. Movies were generated using a filter resolution of 7 Å for solving and visualizing the three components.

helix serves to bridge the kinase, COR and ankyrin domains[29], and deletion of the C-terminal helix in LRRK2 results in insoluble protein[28]. However, in LRRK1, the C-terminal helix does not interact with either the COR or ankyrin domain. Interestingly, a two residue deletion (ΔW1989/ ΔG1990) in the helix in LRRK1 causes OSMD[16]. These two residues sit at the 'base' of the C-terminal helix (Supplementary Fig. 12B). Although we attempted to express this mutant for biochemical characterization, we did not observe any expression, suggesting that even subtle disruption of the C-terminal helix is not tolerated.

### The LRRs of LRRK1 are highly dynamic
Consensus refinements of LRRK1 resulted in a map with a significantly lower local resolution for the LRR domain, suggesting that the LRR is dynamic with respect to the rest of the protein. Recent advances in cryo-EM data processing enable the interrogation of continuous motion from single-particle cryo-EM data[46,47]. To examine the structural dynamics we applied 3D variability analysis[47] in *Cryosparc* to the cryo-EM dataset (Fig. 8, Supplementary Movie 1), solving three variability modes. Briefly, each variability mode corresponds to a vector representing trajectories in the space in which the molecule experiences variability[47]. The first variability component corresponded to a pivoting of the Roc/COR domain/kinase N-lobe relative to the kinase C-lobe and WD40 domain, along with a correlated large, open-and-closing movement of the LRRs (Fig. 8A). The second variability component corresponded to a subtle movement in the Roc domain (Fig. 8B). Finally, the third variability component corresponded to a movement of the Roc and WD40 domain, similar to the first variability component, but without a large movement of the LRRs (Fig. 8C). Similar flexibility in the Roc-COR domain was also observed in the low-resolution cryo-EM map of the C-terminal domain of LRRK1[30]. These structural dynamics

observed in the kinase/COR domains in LRRK1 may reflect inter-domain transitions between active and inactive conformations of the COR and kinase domain. Additionally, the dynamics in the LRR domain may serve to sterically control access to the kinase, with open configurations facilitating substrate binding and the more occluded position sterically hindering substrate binding. Presumably, only the relatively open configurations would allow for substrate binding.

## Discussion
Here, we presented the structure of the full-length LRRK1 monomer, alongside a lower-resolution cryo-EM map of the LRRK1 dimer. Notably, these structures exhibit significant differences to recently determined structures of LRRK2, particularly in the position of the LRRs in the monomer. Furthermore, we show an intrinsically inactive form of the LRRK1 dimer, in which both kinase domains are not only in an inactive conformation but also occluded for substrate binding by the ankyrin repeats from the neighboring molecule. Activation of this dimer would probably require dissociation of the dimer to monomers. It should be noted that our results indicate the presence of both monomer and dimer forms of the full-length LRRK1 in solution. This suggests that the inactivation-activation cycle of the LRRK1 monomer can occur independently, without being a downstream consequence of the dimer-to monomer transition, Further investigations are required to elucidate the factors governing the dimer-to-monomer transition in LRRK1. Moreover, it is crucial to determine the physiological relevance of various oligomerization states observed under cryo-EM conditions for both LRRK1 and LRRK2. Future studies will be necessary to explore these aspects and their implications. We also note a contemporaneous study, where Reimer et al. also reported the structure of LRRK1[48]. Their reported structures are overall very similar to the structures reported here.

One outstanding question that remains to be addressed is the structural reason for the substrate preferences between LRRK1 and LRRK2, specifically the phosphorylation of Rab7A by LRRK1 and Rab8A/Rab10 by LRRK2. Recent work has shown that Rab8/10/12 bind LRRK2 to a conserved binding site in the LRRK2 armadillo domain[31,49,50]. LRRK1 lacks an armadillo domain, so cannot bind Rabs at the N-terminus like LRRK2. It is plausible that the ability of LRRK2 to engage Rabs in the armadillo domain contribute to the distinct biological roles of LRRK1 and LRRK2. However, structures of either LRRK engaged with a substrate Rab in the kinase domain have not yet been reported, and limited information is available regarding the mechanism of LRRK substrate recruitment. Recently, a short sequence in the LRR:Roc domain linker in LRRK2 has been implicated in facilitating the recruitment of large protein substrates, including Rabs, to the LRRK2 kinase domain[51]. Specifically, a tryptophan (W1295 in LRRK2) residue was identified to play a role in substrate recruitment. This tryptophan residue is conserved in LRRK1 (W598 in LRRK1), however the neighboring sequence is markedly different, which may underlie the unique substrate preferences between LRRK1 and LRRK2. However, without experimental data, or structures of Rabs engaged to the LRRK kinase domain, drawing definitive conclusions regarding substrate preferences remain premature.

The structure and biochemical studies presented here highlight the kinase and GTPase interdomain contacts which likely serve to regulate the activity of the LRRK1 kinase. In particular, the extended αC helix, which is unique to LRRK1, serves as a key regulatory scaffold, with mutations in the helix serving to inactivate the kinase through disrupting interactions formed in the kinase active state. The extended αC helix interaction with the COR-B domain likewise serves to stabilize the inactive state of the kinase, disrupting this interface serves to activate the kinase. More distant residues, located at the Roc/COR and in the COR domain more cryptically serve to modulate kinase activity, hinting at the large structural rearrangements, especially involving the DK helix, that must occur to enable LRRK1 kinase activation. Fully understanding this activation mechanism will require experimentally determined structures of LRRK1 in the active state, in particular phosphorylated by its regulatory partners such as PKC. Overall, these results provide a framework for future studies examining LRRK1 biology and enable a comparison of the mechanisms underpinning control of both LRRKs in normal physiology and disease.

## Methods

### Expression and purification of LRRK1

We purchased the full-length human LRRK1 gene from Genscript (residues 1–2015, uniprot Q38SD2), codon optimized for *Homo sapiens*. We sub-cloned the codon-optimized LRRK1 sequence into the pFastBac vector for insect cell protein expression, N-terminally fused to a 3xFlag affinity tag and rhinovirus 3C protease cleavage site, using the NEB HiFi DNA assembly kit (cat. E2621S). We generated bacmid using the Bac-to-Bac system (Invitrogen) and transfected the bacmid into Sf9 cells to generate a baculovirus stock. We subsequently generated a high titer baculovirus stock for protein expression.

4L exponentially growing Sf9 cells (Life Technologies cat. 11496015) at a density of 2–3 × 10$^6$ cells/mL in Sf900-III SFM media (Invitrogen) was infected with the high-titer baculovirus stock. Cells were incubated for 48–72 h at 27 °C with shaking post-infection, and then harvested by centrifugation. The cell pellet was resuspended in resuspension buffer [20 mM Tris, 10 mM CaCl$_2$, 5 mM MgCl$_2$, 100 mM NH$_4$Cl, 100 mM NaCl, 50 mM L-Arg, 50 mM L-Glu, 0.0008% Tween-80, 10% glycerol pH 8.3]. Following resuspension, protease inhibitors (Roche), 10 mM β-glycerophosphate and 1 mM sodium vanadate (to inhibit phosphatases) were added to the pellet, and the pellet was frozen at −80 °C.

All purification steps were done at 4 °C. For purification, the cell pellet was thawed, and cells were lysed using multiple passes through a fluidizer (LM20 Microfluidizer, Microfluidics Corp). The crude lysate was centrifuged, and the supernatant taken and applied to a Flag-M2 affinity column (Sigma cat. A2220) and incubated for 2–3 h. The column was then washed once with resuspension buffer, twice with wash buffer [20 mM Tris, 500 mM NaCl, 5 mM MgCl$_2$, 0.0008% Tween-80 pH 8.3], and then once with gel filtration buffer [20 mM Tris, 150 mM NaCl, 5 mM MgCl$_2$, 0.0008% Tween-80 pH 8.3]. For elution, the column was washed three times with gel filtration buffer supplemented with 150 µg/mL 3×-Flag peptide (Genscript cat. RP21087, Supplementary Fig. 1A).

Eluted fractions containing LRRK1 were concentrated to ~500 µL using a centrifugal concentrator (100 kDa cutoff, Pall cat. MAP100C38) and applied to a Superose 6 Increase 10/30 column (GE Healthcare), equilibrated in gel filtration buffer. The fraction at ~15.5 mL was used for LRRK1 monomer grid preparation (see Supplementary Fig. 1). The fraction at ~14.5 mL was used for LRRK1 dimer grid preparation. Protein was concentrated and used immediately for cryo-EM grid preparation and biochemical/biophysical characterization. The concentration was assessed using UV spectroscopy, using an extinction coefficient of 1.1 cm$^{-1}$(mg/mL)$^{-1}$.

For small-scale affinity purification for kinase assays, mutations were introduced into the full-length, codon-optimized LRRK1 gene using the NEB Q5 Site-Directed Mutagenesis Kit (cat. E0554), and baculovirus was generated as above. Small-scale expressions were done in a volume of 50–250 mL Sf9 cells in Sf900-III media, at a density of 2 × 10$^6$ cells/mL. Cells were infected with baculoviruses, incubated for 60–72 h at 27 °C with shaking post-infection, and then harvested by centrifugation. The cell pellet was frozen once, and then resuspended in lysis buffer [20 mM Tris, 10 mM CaCl$_2$, 5 mM MgCl$_2$, 100 mM NH$_4$Cl, 100 mM NaCl, 50 mM L-Arg, 50 mM L-Glu, 1% Tween-80, 20 µM GppNHP, 10% glycerol pH 8.3], supplemented with protease inhibitors (Roche), 10 mM β-glycerophosphate and 1 mM sodium vanadate (to inhibit phosphatases), and incubated for one hour at 4 °C. Following lysis, the lysate was clarified by centrifugation, and the clarified lysate added to 50 µL Flag-M2 affinity resin (Sigma cat. A2220) and incubated at 4 °C for two hours in 1.5 mL Eppendorf tube. The resin was washed as above, with the exception that 20 µM GppNHP was added to all buffers, and then eluted with a single wash of gel filtration buffer supplemented with 150 µg/mL 3x-Flag peptide (Genscript cat. RP21087) and 20 µM GppNHP. The eluate was filtered using a 0.22 µm Ultrafree-GV centrifugal filter (Sigma cat. UFC30GVNB) to remove any free resin, and then used in a kinase assay. Protein concentration was assessed using UV spectroscopy, using an extinction coefficient of 1.1 cm$^{-1}$(mg/mL)$^{-1}$.

### Mass photometry

Mass photometry data was collected using a OneMP mass photometer (Refeyn). 15 µL of detergent-free gel filtration buffer [20 mM Tris, 150 mM NaCl, 5 mM MgCl$_2$ pH 8.3] was applied to a coverslip, and, after focusing, 3 µL of the undiluted gel filtration fraction was added to the drop and mixed. Movies were acquired for 6,000 frames (60 seconds) using AcquireMP software with the large view setting. Raw data was processed using DiscoverMP software. For calibration, a mixture of beta amylase (Sigma cat. A8781) and thyroglobulin (Sigma cat. T9145) was used.

### Multi-angle light scattering

SEC-MALS data were collected using a Shimadzu LC-20AD HPLC, coupled to a Shimadzu SPD-20A UV detector, Wyatt Dawn MALS detector and Wyatt Optilab refractive index detector. Data were collected following in-line fractionation with a Superose 6 15/150 column (GE Healthcare), pre-equilibrated in gel filtration buffer, running at a flow rate of 0.3 mL/min. 50 µL of the monomer peak from gel filtration chromatography were applied to the column for analysis. Data were analyzed using ASTRA v. 8.0.2.5 (Wyatt). Detector response was

normalized using monomeric BSA (Thermo Fisher, cat. 23209). Protein concentration was determined using differential refractive index, using a dn/dc of 0.184.

## Expression and purification of Rab7A

The sequences for Rab7A (residues 2–176, uniprot P51149) was purchased from Genscript, N-terminally fused to a His tag and tobacco etch virus (TEV) protease cleavage site, codon-optimized, and subcloned into a pET29a vector for expression in *Escherichia coli*. The construct was transformed into sHuffle T7 Express cells[52] (New England Biolabs, cat. C3029J) for protein expression. For protein expression, 6 L of transformed cells in lysogeny broth (LB), supplemented with 50 μg/mL kanamycin, were grown at 37 °C until $OD_{600}$-0.6–0.7, cultures were induced with 1 mM IPTG and grown for 16–18 h at 16 °C, cultures were subsequently harvested by centrifugation, and resuspended in wash buffer [20 mM Tris, pH 7.4, 500 mM NaCl, 20 mM imidazole, 10 mM $MgCl_2$]. Protease inhibitors (Roche) were added to the pellet following harvest, and the pellet was frozen at −80 °C.

All purification steps were done at 4 °C. For purification, the cell pellet was thawed, and cells were lysed using multiple passes through a fluidizer (LM20 Microfluidizer, Microfluidics Corp). The crude lysate was centrifuged, benzonase was added to the lysate (Sigma cat. E1014) and incubated for 15 min at room temperature. The cell lysate was then 0.45 μm filtered and applied to a 5 mL HisTrap HP column (GE Healthcare). The column was washed with five column volumes of wash buffer, then eluted with elution buffer [20 mM Tris, pH 7.4, 500 mM NaCl, 500 mM imidazole, 10 mM $MgCl_2$] over a gradient of twenty column volumes. Fractions containing Rab7A were then pooled. TEV protease was added to the pooled protein (prepared in-house), and then dialyzed overnight against gel filtration buffer [20 mM Tris, pH 7.4, 150 mM NaCl, 10 mM $MgCl_2$]. The following day, the dialysate was taken, and the imidazole and salt concentration adjusted to 20 mM and 500 mM, respectively, and applied to a 5 mL HisTrap HP column to remove TEV protease, free His tags, and uncleaved Rab7A. The flowthrough was taken, which contained Rab7A without a His tag, concentrated to ~5 mL and applied to a Superdex 75 16/160 column (GE Healthcare), equilibrated in gel filtration buffer. Fractions containing purified Rab7A were taken, pooled, concentrated to ~10 mg/mL and frozen at −80 °C for long-term storage. Typical yields were 1–3 mg/L cell culture, with protein concentrations determined using extinction coefficients calculated from the protein sequence.

## LRRK1 kinase activity assays

LRRK1 kinase assays with gel-purified LRRK1 monomer were set up in a 15 μL final mixture with 150 nM LRRK1 monomer, 10 μM Rab7A, 5 mM ATP and 2 mM GppNHP, in 20 mM Tris, 150 mM NaCl, 5 mM $MgCl_2$ and 0.0008% Tween-80. The kinase reaction was carried out at 30 °C for two hours with shaking (300 rpm) in a Thermomixer (Eppendorf). The reaction was stopped by addition of 5 μL of 4× SDS-PAGE loading buffer [250 mM Tris, 8% SDS, 0.2% bromophenol blue, 40% glycerol, 20% β-mercaptoethanol], heated for 10 min at 95 °C, and the samples frozen (−80 °C) prior to immunoblot analysis.

LRRK1 kinase assays with affinity-purified LRRK1 and LRRK1 mutants were set up in 30 μL final mixture with 100 nM LRRK1, 1 μM Rab7A, 5 mM ATP and 2 mM GppNHP, in 20 mM Tris, 150 mM NaCl, 5 mM $MgCl_2$ and 0.0008% Tween-80. The kinase reaction was carried out at 30 °C for 30 min with shaking (300 rpm) in a Thermomixer (Eppendorf). The reaction was stopped by addition of 5 μL of 4× SDS-PAGE loading buffer [250 mM Tris, 8% SDS, 0.2% bromophenol blue, 40% glycerol, 20% β-mercaptoethanol], heated for 10 min at 95 °C, and the samples frozen (−80 °C) prior to immunoblot analysis. Assays were performed from a total of

three independent protein preparations in duplicate (for a total of six independent kinase assays, eighteen assays for the WT and K1270M kinase-dead mutants).

Subsequently, samples were resolved on 4–12% bis–tris gradient gels. Protein was subsequently wet-transferred to a 0.4 μm PVDF membrane. Membranes were subsequently blocked for 30 min using 5% bovine serum albumin (BSA) in TBS-T [20 mM Tris, 150 mM NaCl pH 7.4, 1% Tween-20] and probed with anti-phospho Rab7A primary antibody (Abcam cat. ab302494, diluted 1:1000), Rab7A (total) primary antibody (Abcam cat. ab50533, diluted 1:2000) and anti-LRRK1 primary antibody (Abcam cat. ab228666, diluted 1:2000). Membranes were washed using TBS-T and then probed with goat anti-rabbit IR-fluorescent secondary antibody (LiCor cat. 926-3221, diluted 1:20,000) and goat anti-mouse IR-fluorescent secondary antibody (LiCor cat. 926-68072). The membrane was subsequently washed with TBS-T and imaged using a Typhoon scanner (GE Healthcare, software v. 1.1.0.7). Blots were quantified using Fiji[53] to determine the pRab7A/Rab7A ratio, this ratio was then normalized to LRRK1 WT reactions on the same membrane. Significance of differences was quantified using a one-way Brown-Forsythe and Welch ANOVA test in Prism 9.4.1, using an unpaired t-test with Welch's correction to compare mutants with the WT and K1270M kinase-dead mutant.

## Cryo-EM – grid preparation, data collection and 3D reconstruction

Cryo-EM was performed at the cryo-EM facility in the Center for Structural Biology, NCI-Frederick. 1.5 μL purified LRRK1 monomer or dimer at a concentration of 0.2–0.5 mg/mL was applied to each side of a Quantifoil R 1.2/1.3 Gold 300 Mesh grids (cat. Q3100AR1.3), that had been glow discharged on each side (25 mA for 30 s). Grids were vitrified using a Leica EM GP2 plunge freezer, with a blotting time of 1–3 s. Grids were subsequently imaged using a Gatan K3 direct detector, equipped with an energy filter, mounted on a Talos Arctica G2 (Thermo Fisher) electron microscope in super resolution mode (pixel size 0.405 Å/pixel, ×100,000 magnification). 50 frames per movie were acquired for a total dose of approximately 50 electrons/Å$^{-2}$. Data was collected using the EPU program (Thermo Fisher), with defocus values ranging from −2.5 to −0.8 μm. 22,865 movies were collected from LRRK1 monomer grids, and 9992 movies from LRRK1 dimer grids, for a total of 32,157 movies.

Data processing was performed using Cryosparc 3.3[54]. Movies were imported into Cryosparc, patch-motion and patch-CTF corrected. Movies were binned to the physical pixel size in the patch motion step. Movies with a CTF resolution >5 Å were removed from the stack, leaving 18,074 monomer movies and 8612 dimer movies for further processing (26,686 movies total). 3,553,719 particles were picked from the monomer movies and 1,133,944 from the dimer movies (for a total of 4,687,663 particles), using a Topaz model trained on an initial LRRK1 monomer or dimer dataset[55,56]. Particle stacks were subsequently merged and curated using two rounds of 2D classification to remove clear false positive particles, carbon edges and junk particles, with duplicate particles which were introduced due to re-centering removed after each round of 2D classification. The 517,408 particles were used in a five-class ab initio model generation in Cryosparc. Separately, 26,352 particles corresponding to the LRRK1 dimer were used to generate an initial model of the LRRK1 dimer (representative classes used to generate the dimer volume indicated with an asterisk in Supplementary Fig. 2B). These models were used as the input for a seven-class heterogenous refinement with the full stack of 517,408 particles (Supplementary Fig. 2). Particles from the dimer class were used in a subsequent non-uniform refinement[57] step, giving a map with an overall resolution of 6.38 Å, judged by

the gold standard FSC in Cryosparc (Supplementary Fig. 2). Particles from monomer classes were used in a second heterogenous refinement round, with four classes, with the 183,273 particles from the best class used in a final non-uniform refinement step, giving a map with an overall resolution of 3.92 Å, judged by the gold standard FSC in Cryosparc (Supplementary Fig. 2). Subsequently, 3D variability analysis[47] was used in Cryosparc to analyze the structural dynamics of LRRK1. Three variability modes were solved, using a filter resolution of 7 Å, and 3D variability was visualized using 'intermediates' mode, with ten frames and a filter resolution of 7 Å.

Inspection of the map revealed that the LRR region was poorly defined relative to the rest of the map. To improve the resolution of the C-terminal region, the LRR region was subtracted from the monomer map using Particle Subtraction in Cryosparc. Local refinement in Cryosparc, using a soft mask corresponding to the C-terminal region, was then used to improve the resolution of the C-terminal region to 3.78 Å. Local resolution maps and FSC curves were generated using Cryosparc. For a graphical summary of the cryo-EM image processing, see Supplementary Fig. 3. For a summary of cryo-EM reconstruction statistics, see Supplementary Table 1. We combined the Cryosparc-sharpened global and local refinement maps of the monomer using the *phenix.combine_focused_maps* in Phenix 1.20[58] for atomic model refinement. The global and locally refined monomer maps were additionally post-processed using *deepEMhancer* v 0.13[59], these maps were deposited in the EMDB deposition and used to prepare some of the figures in place of the combined map from *Phenix*. The map used (composite map, map sharpened in *Cryosparc* or map post-processed using *deepEMhancer*) is indicated in the figure legend.

### Cryo-EM−atomic model refinement

For atomic model building and refinement, we first pre-processed the Alphafold[39,60] predicted model of LRRK1 (AF-Q38SD2-F1) using *phenix.process_predicted_model* in Phenix 1.20[58], which removed low-confidence regions in the predicted model, and split the model into domains for subsequent rigid body fitting. We fit the domains into the map using *UCSF Chimera*[61] to generate an initial model. We refined this model against the composite cryo-EM map using *phenix.real_space_refine*[62], followed by manual model building in *Coot*[63] and subsequent refinement in *phenix.real_space_refine*. Geometry and real-space correlation validation was performed using the *phenix.validation_cryo-EM* tool (incorporating *MOLProbity*[64]). Figures were prepared using *UCSF ChimeraX*[65,66]. For a summary of model building and validation statistics, see Supplementary Table 1.

For the LRRK1 dimer atomic model, two copies of the LRRK1 monomer were docked into the map using *UCSF Chimera*, and the ankyrin repeats from the Alphafold model of LRRK1 were added. This model was subsequently refined using *phenix.real_space_refine* using five rounds of rigid body refinement, defining the ANK repeats, LRRs, Roc-COR and kinase-WD40 domains as separate rigid bodies. Due to the low resolution of the map, we did not deposit an atomic model in the PDB.

### Differential scanning fluorometry

Thermal stability was measured using the Prometheus NT.48 nano-DSF instrument (NanoTemper). LRRK1 or the relevant mutant in the buffer 20 mM Tris, 150 mM NaCl, 5 mM MgCl$_2$, 0.0008% Tween-80 pH 8.3, at a concentration of 0.1 mg/mL was loaded into a nanoDSF glass capillary. Thermal unfolding was measured at a heating rate at 1 °C, protein melting temperature was calculated from the first derivative of the ratio of tryptophan fluorescence at 330 nm and 350 nm.

### Cell culture, transfection and SDS-PAGE analysis of in vitro kinase activity

U2OS cells (ATCC, cat #HTB-96) were maintained in DMEM containing 4.5 g/l glucose, 2 mM l-glutamine, 1% Pen/Strep and 10% FBS at 37 °C in 5% CO$_2$. Cells were seeded on 12 mm coverslips pre-coated for immunostaining with Matrigel (Corning). For western blot, cells were seeded on 12-well plates.

Transient transfections of U2OS cells were performed using Lipofectamine Stem Reagent (ThermoFisher, cat #STEM00015) and incubated for 48 hours before fixation and lysis. U2OS cells were transfected with 3xFLAG-LRRK1 (ICC and WB) and GFP-Rab7 (ICC) for 48 h.

Proteins were resolved on 4−20% Criterion TGX precast gels (Biorad, cat #5671095) running at 200 V for 40 min. Gels were then transferred to nitrocellulose membranes (Biorad, cat #170415) by semi-dry trans-Blot Turbo transfer system (Biorad). The membranes were blocked with Odyssey Blocking Buffer (Licor, cat #927-40000) and then incubated overnight at 4 °C with primary antibodies. The primary antibodies used were mouse anti-FLAG (1:10,000, Millipore-Sigma cat#F3165), rabbit anti-pS72-Rab7 (1:1000; Abcam, cat#ab302494), mouse anti-Rab7 (1:2000, Cell Signaling Technologies, cat#95746) and mouse anti-αtubulin (1:10,000, Cell Signaling Technologies, cat#3876). Afterward, membranes were washed in TBST (3 × 5 min) followed by incubation for 1 hour at RT with fluorescently conjugated secondary antibodies as previously stated above. The blots were washed in TBST (3 × 5 min) and scanned on an ODYSSEY® CLx. Quantitation of bands was performed using Image Studio (Licor). All blots were probed for α-tubulin, used as a housekeeping protein to ensure equal loading of samples.

### Confocal microscopy and analyses

U2OS cells were fixed with 4% PFA for 10 min, permeabilized with PBS/0.1% Triton X-100 for 10 min, and blocked with 5% donkey serum for 1 h at RT. Mouse anti-FLAG (M2, 1:500, Millipore-Sigma cat#F3165) was diluted in blocking buffer (1% donkey serum) and incubated overnight at 4 °C. After three 5-min washes with PBS/0.1% Triton X-100, donkey anti-mouse secondary fluorescently labeled antibody (1:500, Alexa Fluor™ 568, Thermo Fisher Scientific cat#A10037) was diluted in blocking buffer (1% donkey serum) and incubated for 1 hour at RT. Coverslips were washed twice with 1× PBS and an additional two times with dH$_2$O and mounted with ProLong Gold antifade reagent (Thermo Fisher Scientific).

Spinning disk super-resolution microscopy was used on a W1-SoRa super-resolution spinning disk microscope (Nikon) with a 60 × 1.49 NA oil immersion objective. A 4× intermediate magnification (240× combined) was used. For deconvolution, we used 20−25 iterations of the 3D Landweber algorithm with the NIS-Elements AR 5.21.03 software. Images were acquired simultaneously using two Photometrics prime 95b sCMOS cameras, a 565LP DM, and appropriate emission cleanup filters. Triggered piezo was used to maximize speed. Stacks were taken with 0.2 µm distance between slices. Only low plasmid expressing cells without obvious overexpression artifacts were imaged. For measuring colocalization, Fiji plugin. Colocalization Threshold was used to analyze Pearson's correlation to quantify LRRK1:Rab7 colocalization (ImageJ, NIH). Cells with an R < threshold of >0.2 were discarded from the analysis.

### Reporting summary

Further information on research design is available in the Nature Portfolio Reporting Summary linked to this article.

## Data availability

The cryo-EM maps generated in this study have been deposited in the EMDB database under accession code EMD-28950 (LRRK1 monomer

composite map), EMD-28949 (LRRK1 monomer global refinement), EMD-28951 (LRRK1 C-terminus local refinement after subtraction of the LRRs), and EMD-28952 (LRRK1 dimer). The atomic model of the LRRK1 monomer has been deposited in the PDB with accession code 8FAC. The quantified kinase assay, confocal microscopy and differential scanning fluorimetry melting temperature data generated in this study are provided in the Source Data file. Uncropped blots and gels are presented in the Supplementary Information. Source data are provided with this paper.

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

## Acknowledgements

We thank Dr. Dan Shi for assistance collecting the cryo-EM data, Dr. Sergey G. Tarasov and Marzena Dyba for assistance with collecting the differential scanning fluorometry and mass photometry data. Cryo-EM data was collected using the Talos Arctica G2 in the Center for Structural Biology cryo-EM facility, NCI at Frederick. We acknowledge use of the Biophysics Resource, Center for Structural Biology, NCI at Frederick, and use of the Frederick Research Computing Environment cluster. This research was supported by federal funds from the intramural program of the National Cancer Institute, National Institutes of Health, under project number ZIA BC 011744 (P.Z.), and was additionally supported in part by the Intramural research Program of the National institute on Aging, National Institutes of Health.

## Author contributions

R.D.M., J.A.M.F and P.Z. conceptualized and initiated the project. R.D.M. and J.A.M.F. collected cryo-EM data, R.D.M. processed the cryo-EM data with input from J.A.M.F. and P.Z, R.D.M. prepared all recombinant protein samples and undertook the biochemical/biophysical characterization. L.B-P and J.H.K. performed the in vitro cell assays and super-resolution microscopy. M.R.C and P.Z. supervised the work. R.D.M. wrote the first draft of the manuscript, all authors approved the final version of the manuscript.

## Funding

## Competing interests

The authors declare no competing interests.
