## [Peer review file · Nature Communications]

REVIEWER COMMENTS

Reviewer #1 (Remarks to the Author):

The study by Zhang P and colleagues presented the full-length LRRK1 structure, the key points of the paper include:

- structure of LRRK1 in both monomer (3.9A) and dimer (6.4A) forms
- structural comparison between LRRK1 and LRRK2
- structural and functional characterization of the aC-DK helix interactions, ROC-COR interactions, and kinase-WD40 interactions.
- the flexible nature of the LRR domain.

Overall, the work provided the cryoEM structure of human LRRK1 and showed key intramolecular interactions that are critical for the function, making it a suitable candidate for Nat. Comm. after revision.

Major points:

1. For final structural model validation, please provide a Fourier shell coefficient (FSC) curve between the model and map to show the model quality.
2. For structural comparison between LRRK1 and LRRK2, please provide a figure with an overlay of the catalytic C-terminal halves with RMSD.
3. For Fig. 3-5, it would be helpful to show readers zoom-out views of the overall structure, indicating which part of the interactions the authors are talking about.
4. When testing the aC-DK helix interaction, could the author do the M1298K/R mutation to test if it could stabilize the inactive state?
5. Can authors provide insights or comments on potential conformational changes needed for kinase activation by comparing with the AF2 model (presumably an active conformation)? Meanwhile, could the authors rationalize which interfaces are important for stabilizing the inactive state based on the mutagenesis analysis and structural comparison?

Minor points:

1. typo: "the overall arraignment of the C-terminal catalytic"
2. "and thus does not form extensive electrostatic... this position of the helix (Figure 3D)"

Here Figure 3D does not seem to support the statement.

3. typo: "for controlling the LRRK1 inactive to active transition"

Reviewer #2 (Remarks to the Author):

The manuscript by Metcalfe and colleagues entitled "Structure and regulation of full-length human leucine-rich repeat kinase 1" investigates the 3D structure of the ROCO protein LRRK1 using cryo-EM. This is a thoroughly conceived and executed study that uncovers the structural basis of LRRK1, comparing and contrasting the kinase with its homologous LRRK2.

I have the following remarks for the authors:

1) LRRK1 phosphorylates Rab7A - but not Rab8A/10 – and LRRK2 phosphorylates Rab8/10 but not Rab7A at the equivalent sites. Can the authors make some structural predictions to explain this specificity?

2) Hanafusa and colleagues showed that LRRK1 is phosphorylated by PLK1 at S1817 and by CDK11 at S1427, which lies in the activation loop. The authors should assess the structural effects of these phosphorylations and discussed them in light of the reported biological effect (PMIDs: 36254578, 26192437)

Reviewer #3 (Remarks to the Author):

The authors describe two new cryo-EM structures of LRRK1 monomer and dimer. The monomer structure is resolved at higher resolution and the dimer at a low resolution. The authors also report additional biochemical results to validate their hypotheses about functional aspects of the monomer structure. They also report processing results using 3D Variability Analysis to show the flexibility and conformational distribution of LRRK1.

I was asked to comment specifically on the data processing aspects and analysis of structural variability in this work, so I will not comment on the biological or other experimental aspects, though from my

reading it does appear that the paper is sound and conclusions are reasonable given the data. It is also well written and clear.

Overall, I feel that the data analysis and structural variability analysis in this work are sound, and I do not have any major concerns. I only have a few minor questions and suggestions for clarification in the manuscript. I would happily recommend this manuscript for publication.

First, some noteworthy aspects of the data analysis:

- * The monomer and dimer fractions were imaged separately on separate grids, and the data from these was combined during processing. This is reasonable given that there were many monomer particles in the dimer fraction.
- * The authors note that the 3.9/3.8Å result they have for the monomer structure is reasonable given the details that appear in the density map, I would agree.
- * The authors also explain that the dimer is resolved at a resolution where docking of major subunits of the monomer is appropriate (and they see density for the ankyrin repeats that is not visible in the monomer) but that the dimerization interface cannot be analyzed in detail. I agree that this is appropriate.
- * 3D Variability is used to interrogate continuous motion and flexibility that appears to be present in the LRRK1 monomer.

Questions and suggestions:

1) The 3DVA results are used to draw some (tentative) conclusions in the paper, and these seem appropriate. There is one missing piece of information about the 3DVA results however - at what filter resolution was 3DVA run? The 3DVA algorithm's results depend strongly on the resolution used during solving the variability components; for larger motions, lower resolutions are needed and for smaller, fine motions, higher resolutions are appropriate. Given the 3DVA supplementary movie and Figure 6, it appears that 3DVA is finding large motions of the LRR/ANK and also coordinated bending of the overall structure. For these large motions, a medium/low resolution would be appropriate. I do not have any concern that the results presented are not correct, but two suggestions: i) the authors should mention the filter resolution used during solving the components in the methods section if not in the main paper where Figure 6 is discussed and ii) if possible, the authors should run 3DVA with a somewhat lower filter resolution than what was used for Figure 6. This will help the results be more interpretable and "cleaner" (as currently there is a fair bit of noise in the movies). This is optional and not necessary for the (not overly strong) conclusions that are drawn from the 3DVA results. The methods section does mention that a filter resolution of 4Å was used during visualization of the 3DVA results, but not the resolution that was used during solving of the components.

2) In the main processing workflow, it is notable that two datasets that were prepared differently were merged at the early stages of processing. The authors indicate in the text that dimer particles were separated from the full set based on 2D classification. This implies that dimer particles were easily distinguishable in 2D. Supp Figure 2 shows 2D classes and from these it is obvious that the “front” view of the dimer is obviously identifiable in the 2D classes. However it is not clear that the top/side views of the dimer would be immediately recognizable given they have nearly the same dimensions as a top/side view of the monomer. This is a minor point, since in the processing flow, all the particles are combined into a single heterogeneous refinement and therefore all particles have a chance to switch to either monomer or dimer classes. However as a suggestion to improve the manuscript, I would suggest that the authors i) add a sentence or two to the methods section to clarify any specific details of how the data were combined/separated, and ii) in Supp figure 2, label the classes in B that were manually selected as dimers.

Response to Reviewers:

We would like to thank the reviewers for their supportive comments and helpful suggestions. Our specific responses to reviewer comments are outlined below.

Review comments are highlighted in blue, our responses are provided in black.

Reviewer #1 (Remarks to the Author):

The study by Zhang P and colleagues presented the full-length LRRK1 structure, the key points of the paper include:

- structure of LRRK1 in both monomer (3.9A) and dimer (6.4A) forms
- structural comparison between LRRK1 and LRRK2
- structural and functional characterization of the aC-DK helix interactions, ROC-COR interactions, and kinase-WD40 interactions.
- the flexible nature of the LRR domain.

Overall, the work provided the cryoEM structure of human LRRK1 and showed key intramolecular interactions that are critical for the function, making it a suitable candidate for Nat. Comm. after revision.

Major points:

1. For final structural model validation, please provide a Fourier shell coefficient (FSC) curve between the model and map to show the model quality.

We have included the map-model FSC for the LRRK1 monomer (Supplementary Figure 3C). We did not add a map-model FSC curve for the LRRK1 dimer, as we did not deposit an atomic model of the LRRK1 dimer.

2. For structural comparison between LRRK1 and LRRK2, please provide a figure with an overlay of the catalytic C-terminal halves with RMSD.

We have included additional panels in Figure 2 (Figure 2D) showing an overlay of the catalytic C-terminal Roc-Cor, kinase, WD40 ('RCKW') domains of LRRK1 and LRRK2.

3. For Fig. 3-5, it would be helpful to show readers zoom-out views of the overall structure, indicating which part of the interactions the authors are talking about.

We thank the reviewer for this suggestion. A 'zoomed-out' view has been added to Figures 3-5 (and Figure 6) to aid the reader.

4. When testing the aC-DK helix interaction, could the author do the M1298K/R mutation to test if it could stabilize the inactive state?

We thank the reviewer for this suggestion, as the M1298 mutation is indeed an intriguing one to investigate. The proximity of M1298 in the α C helix to several negatively charged residues in the COR-B DK helix (e.g., S1139, D1135) makes it interesting to replace M1298 with a positively charged R/K residue, which could potentially form a new interaction with the DK helix. This interaction may stabilize the inactive position of both the DK and α C helices.

To explore the effects of the M1298K mutation, we performed experiments both *in vitro* using affinity purified LRRK1. Surprisingly, the M1298K mutation resulted in an ~4 fold increase in kinase activity compared to both the WT kinase and the M1298A mutation (see Figure 4 in the resubmitted manuscript). In our revised manuscript, we provide a speculative explanation for these findings by proposing the creation of a new interaction between the α C helix and the DK helix, which potentially stabilizes the interaction in its 'active' state (see lines 265-271). However, as with our response to point 5 below, we would like to emphasize that these explanations are speculative and preliminary in the absence of experimental structural data.

We also note here that during the generation of the new mutants characterized in this resubmission, we noticed an error in the initial manuscript, where the M1288A mutation was incorrectly referred to as M1298A. We have addressed this mistake, generated, and provided a detailed discussion of the correct M1298A mutation (lines 262-269), alongside the M1298K mutant, and an M1288L mutation (lines 300-302).

5. Can authors provide insights or comments on potential conformational changes needed for kinase activation by comparing with the AF2 model (presumably an active conformation)? Meanwhile, could the authors rationalize which interfaces are important for stabilizing the inactive state based on the mutagenesis analysis and structural comparison?

The AlphaFold^{1,2} model of LRRK1 (AF_Q38SD2) does have the kinase domain in an active conformation, which is consistent with typical AlphaFold predictions of kinase structures. Notably, the interactions between the COR-B domain and the kinase domain in the AlphaFold prediction closely resemble the interactions in the experimental cryo-EM structure of LRRK2 in the active conformation³. However, we note at the time of writing (June 2023), the coordinates of the experimental structure of LRRK2 in the active conformation are not publicly released, which prevent us from thoroughly interrogating the difference between the inactive state of LRRK1/2 and the active state of LRRK2, as well as comparing the AlphaFold prediction of the active state of LRRK1 with the experimentally determined structure of the LRRK2 active state. We have expanded on our discussion of the consequences of several of the mutations on the activity of LRRK1 by incorporating the AlphaFold model in the resubmitted manuscript (for e.g. lines 223-233, Figure 4D).

Regarding the inter-domain interactions in the AlphaFold predicted active state of LRRK2, the positions of the catalytic Roc-COR, kinase, WD40 ('RCKW') domains are similar between the experimental inactive-state structure and the predicted active-state structure. Additionally, the predicted alignment error (PAE) from AlphaFold for residues in this region of the protein suggests that the arrangement of the RCKW domains is accurate. However, the PAE is high

(indicating inaccuracy) for the alignment between the RCKW domains, the LRRs and the ANK repeats. Consequently, no definite conclusions can be drawn regarding the position of these domains in the active state of LRRK1. Clearly, a comprehensive understanding of any large rearrangements that occur upon LRRK1 activation requires experimental structural studies of active LRRK1.

Minor points:

1. typo: "the overall arraignment of the C-terminal catalytic"
2. "and thus does not form extensive electrostatic... this position of the helix (Figure 3D)"
Here Figure 3D does not seem to support the statement.
3. typo: "for controlling the LRKR1 inactive to active transition"

We thank the reviewer for these corrections. They have been made to the manuscript.

Reviewer #2 (Remarks to the Author):

The manuscript by Metcalfe and colleagues entitled "Structure and regulation of full-length human leucine-rich repeat kinase 1" investigates the 3D structure of the ROCO protein LRRK1 using cryo-EM. This is a thoroughly conceived and executed study that uncovers the structural basis of LRRK1, comparing and contrasting the kinase with its homologous LRRK2.

I have the following remarks for the authors:

1) LRRK1 phosphorylates Rab7A - but not Rab8A/10 – and LRRK2 phosphorylates Rab8/10 but not Rab7A at the equivalent sites. Can the authors make some structural predictions to explain this specificity?

We thank the reviewer for raising this intriguing question. It is a significant and broad question that falls outside the experimental scope of our manuscript. However, in light of recent findings, we can offer some speculative insights.

We note three recent manuscripts that examine the interaction between LRRK2 and Rab8/10/12/29³⁻⁵. These studies conclude that Rab8/10/29 bind to a shared binding site in the armadillo domain, while Rab12 binding to a neighboring but distinct site. Since LRRK1 lacks an armadillo domain, it lacks the ability to bind Rabs at the extreme N-terminus in a similar manner to LRRK2. This distinction in binding capacity could be one of the factors contributing to the distinct biological roles played by LRRK1 and LRRK2. However, it is important to note that these studies do not address the mechanisms by which LRRK2 engages its substrate Rabs via the kinase domain and/or how they would be translocated to the kinase domain for phosphorylation.

We would also like to highlight a recent paper by Weng *et al.*⁶ that identified a short linker sequence in the LRRK2 LRR:Roc domain linker, which is involved in the phosphorylation of large protein substrates, such as Rabs, but not small peptide substrates. This region is located on the

final leucine-rich repeat near the Roc domain, and the structure alignment between LRRK1 and LRRK2 diverges (Review Figure 1A-B) just before this region. Interestingly, while the key residue W1295 involved in protein substrate recruitment is conserved in LRRK1 (W598), there are marked differences in the sequence of the linker region and neighboring residues (Review Figure 1C). Moreover, this region of the LRRs forms direct contacts with the kinase domain in LRRK2, which is not present in LRRK1, due to the differing position of the LRRs in the two proteins. Thus, the sequence differences, along with corresponding variations in surface charge and hydrophobicity in this region, as well as the differences between the LRR/kinase interaction, may contribute to different substrate preferences between the two LRRKs.

In response to this question, we have added some text to the conclusion section of the manuscript (lines 472-487) to address these points. Nevertheless, we emphasize (here and in the manuscript) that in the absence of experimental structural data, any conclusions drawn remain preliminary and speculative.

Review Figure 1: Differences in the LRR/Roc linker and LRR/kinase interactions between LRRK1 and LRRK2. A) Overall structural alignment between LRRK1 and LRRK2 (PDB: 7LHW⁷, aligned on kinase domain). B) Detail of the LRR/kinase domain interactions in i), LRRK1 and ii), LRRK2. The region detailed is highlighted in panel A) with a box, domains are colored as in Figure 1 in the manuscript. Residue W598 (LRRK1)/W1295 (LRRK2) and the LRR/kinase interactions in LRRK2 are highlighted. C) Sequence alignment of the region around tryptophan 598 (LRRK1)/1295 (LRRK2).

2) Hanafusa and colleagues showed that LRRK1 is phosphorylated by PLK1 at S1817 and by CDK11 at S1427, which lies in the activation loop. The authors should assess the structural effects of these phosphorylations and discussed them in light of the reported biological effect (PMIDs: 36254578,)

We thank the reviewer for these comments. In response, we have incorporated a comment regarding these residues into the text, and we have cited both relevant papers (lines 381-386).

Notably, both S1817 and T1427 are not resolved in the structure. Phosphorylation of T1427 activates LRRK1, and the T1427A mutation is intrinsically inactive. As it sits in the activation loop, phosphorylation of this residue will likely result in the stabilization of the kinase active state, analogous to the well-studied kinase PKA⁸.

Regarding S1817, it is also unresolved in the structure and sits in a long loop in the WD40 domain which is also predicted as disordered in the AlphaFold model of the LRRK1 structure. Phosphorylation of this serine may contribute to altering the interaction with other interacting partners, thereby modulating LRRK1 activity.

Reviewer #3 (Remarks to the Author):

The authors describe two new cryo-EM structures of LRRK1 monomer and dimer. The monomer structure is resolved at higher resolution and the dimer at a low resolution. The authors also report additional biochemical results to validate their hypotheses about functional aspects of the monomer structure. They also report processing results using 3D Variability Analysis to show the flexibility and conformational distribution of LRRK1.

I was asked to comment specifically on the data processing aspects and analysis of structural variability in this work, so I will not comment on the biological or other experimental aspects, though from my reading it does appear that the paper is sound and conclusions are reasonable given the data. It is also well written and clear.

Overall, I feel that the data analysis and structural variability analysis in this work are sound, and I do not have any major concerns. I only have a few minor questions and suggestions for clarification in the manuscript. I would happily recommend this manuscript for publication.

First, some noteworthy aspects of the data analysis:

- * The monomer and dimer fractions were imaged separately on separate grids, and the data from these was combined during processing. This is reasonable given that there were many monomer particles in the dimer fraction.
- * The authors note that the 3.9/3.8Å result they have for the monomer structure is reasonable given the details that appear in the density map, I would agree.
- * The authors also explain that the dimer is resolved at a resolution where docking of major subunits of the monomer is appropriate (and they see density for the ankyrin repeats that is not visible in the monomer) but that the dimerization interface cannot be analyzed in detail. I agree that this is appropriate.
- * 3D Variability is used to interrogate continuous motion and flexibility that appears to be present in the LRRK1 monomer.

Questions and suggestions:

1) The 3DVA results are used to draw some (tentative) conclusions in the paper, and these seem appropriate. There is one missing piece of information about the 3DVA results however - at what filter resolution was 3DVA run? The 3DVA algorithm's results depend strongly on the resolution used during solving the variability components; for larger motions, lower resolutions are needed and for smaller, fine motions, higher resolutions are appropriate. Given the 3DVA supplementary movie and Figure 6, it appears that 3DVA is finding large motions of the LRR/ANK and also coordinated bending of the overall structure. For these large motions, a medium/low resolution would be appropriate. I do not have any concern that the results presented are not correct, but two suggestions: i) the authors should mention the filter resolution used during solving the components in the methods section if not in the main paper where Figure 6 is discussed and ii) if possible, the authors should run 3DVA with a somewhat lower filter resolution than what was used for Figure 6. This will help the results be more interpretable and "cleaner" (as currently there is a fair bit of noise in the movies). This is optional and not necessary for the (not overly strong) conclusions that are drawn from the 3DVA results. The methods section does mention that a filter resolution of 4Å was used during visualization of the 3DVA results, but not the resolution that was used during solving of the components.

We thank the reviewer for this comment. In the initial manuscript, we used a filter resolution of 4 Å for both the visualization of the 3DVA results and the initial solving of the components.

To facilitate a direct comparison, we performed 3DVA using a filter resolution of 5 Å and 7 Å for solving of the components and the visualization of the results. These results are presented in Review Figure 2 (for comparison, alongside the results previously presented in Figure 6). We agree with the reviewer that the results are indeed 'smoother' and have updated Figure 8 (previously Figure 6) and the Supplementary Movie. Moreover, we have updated the Methods to explicitly state the filter resolution used (line 684-685) for both solving the components and visualizing the 3DVA results and mentioned it in the caption for Figure 8.

Review Figure 2: Comparison of 3DVA results for LRRK1 solved using, A) a filter radius of 4 Å (presented in Figure 6 in the initially submitted manuscript), B) a filter radius of 5 Å and C) a filter radius of 7 Å (presented in Figure 8 in the resubmitted manuscript).

2) In the main processing workflow, it is notable that two datasets that were prepared differently were merged at the early stages of processing. The authors indicate in the text that dimer particles were separated from the full set based on 2D classification. This implies that dimer particles were easily distinguishable in 2D. Supp Figure 2 shows 2D classes and from these it is obvious that the “front” view of the dimer is obviously identifiable in the 2D classes. However it is not clear that the top/side views of the dimer would be immediately recognizable given they have nearly the same dimensions as a top/side view of the monomer. This is a minor point, since in the processing flow, all the particles are combined into a single heterogeneous refinement and therefore all particles have a chance to switch to either monomer or dimer classes. However as a suggestion to improve the manuscript, I would suggest that the authors i) add a sentence or two to the methods section to clarify any specific details of how the data were combined/separated, and ii) in Supp figure 2, label the classes in B that were manually selected as dimers.

We thank the reviewer for this comment. They have correctly appraised our data processing approach. To elaborate, in our data processing strategy we initially generated a clear ‘dimer’ volume from the data for subsequent classification of dimer particles in the subsequent heterogeneous refinement step. As the reviewer notes, ‘side’ views of the dimer are virtually indistinguishable by eye from side views of the monomer. Therefore, we used only the top views of the dimer particles to generate the initial dimer volume, as these particles were unambiguously dimer particles. The subsequent heterogeneous refinement step allowed us to classify the full particle stack into monomer, dimer, and junk particles. It is important to note that when we exclusively use particles from the ‘monomer’ grids to generate a monomer reconstruction, we observed modestly lower resolution (4.1 Å) compared to using monomer particles from both the monomer and dimer grids.

To address this in more detail, we have expanded the corresponding section in the methods section (lines 673-676). Additionally, we have marked representative ‘dimer’ views used to generate the initial dimer volume with an asterisk in Supplementary Figure 2B.

References:

- 1 Jumper, J. *et al.* Highly accurate protein structure prediction with AlphaFold. *Nature* **596**, 583-589, doi:10.1038/s41586-021-03819-2 (2021).
- 2 Varadi, M. *et al.* AlphaFold Protein Structure Database: Massively expanding the structural coverage of protein-sequence space with high-accuracy models. *Nucleic Acids Research* **50**, D439-D444, doi:10.1093/nar/gkab1061 (2022).
- 3 Zhu, H., Tonelli, F., Alessi, D. R. & Sun, J. Structural basis of human LRRK2 membrane recruitment and activation. *bioRxiv*, 2022.2004.2026.489605 (2022).

- 4 Dhekne, H. S. *et al.* Genome-wide screen reveals Rab12 GTPase as a critical activator of pathogenic LRRK2 kinase. *bioRxiv* (2023).
- 5 Vides, E. G. *et al.* A feed-forward pathway drives LRRK2 kinase membrane recruitment and activation. *eLife* **11**, 1-29, doi:10.7554/eLife.79771 (2022).
- 6 Weng, J.-H. *et al.* Novel LRR-ROC motif that links the N- and C-terminal Domains in LRRK2 undergoes an order-disorder transition upon activation. *Journal of Molecular Biology* **In press**, doi:10.1016/j.jmb.2023.167999 (2023).
- 7 Myasnikov, A. *et al.* Structural analysis of the full-length human LRRK2. *Cell*, 1-9, doi:10.1016/j.cell.2021.05.004 (2021).
- 8 Steichen, J. M. *et al.* Structural basis for the regulation of protein kinase A by activation loop phosphorylation. *Journal of Biological Chemistry* **287**, 14672-14680, doi:10.1074/jbc.M111.335091 (2012).